# Engineering Hydrogels for the Development of Three-Dimensional In Vitro Models

**DOI:** 10.3390/ijms23052662

**Published:** 2022-02-28

**Authors:** Somnath Maji, Hyungseok Lee

**Affiliations:** 1Department of Mechanical and Biomedical Engineering, Kangwon National University (KNU), Chuncheon 24341, Korea; somnath.2812@gmail.com; 2Department of Smart Health Science and Technology, Kangwon National University (KNU), Chuncheon 24341, Korea

**Keywords:** in vitro model, hybrid hydrogel, extracellular matrix, microenvironment mimicking matrix

## Abstract

The superiority of in vitro 3D cultures over conventional 2D cell cultures is well recognized by the scientific community for its relevance in mimicking the native tissue architecture and functionality. The recent paradigm shift in the field of tissue engineering toward the development of 3D in vitro models can be realized with its myriad of applications, including drug screening, developing alternative diagnostics, and regenerative medicine. Hydrogels are considered the most suitable biomaterial for developing an in vitro model owing to their similarity in features to the extracellular microenvironment of native tissue. In this review article, recent progress in the use of hydrogel-based biomaterial for the development of 3D in vitro biomimetic tissue models is highlighted. Discussions of hydrogel sources and the latest hybrid system with different combinations of biopolymers are also presented. The hydrogel crosslinking mechanism and design consideration are summarized, followed by different types of available hydrogel module systems along with recent microfabrication technologies. We also present the latest developments in engineering hydrogel-based 3D in vitro models targeting specific tissues. Finally, we discuss the challenges surrounding current in vitro platforms and 3D models in the light of future perspectives for an improved biomimetic in vitro organ system.

## 1. Introduction

In a living body, cells are enclosed firmly in a three-dimensional (3D) mass of matrix, where they are constantly proliferating, migrating, differentiating, and communicating with each other and their immediate microenvironment. This area that surrounds the cells is an intricate network of multi-domain macromolecules and other biological factors organized in a cell- and tissue-specific manner, and is termed as extracellular matrix (ECM) [1]. Faithfully replicating such a complex microenvironment of any tissue in in vitro cell culture conditions is still a challenge for the scientific community. For a very long time, two-dimensional (2D) cell culture was commonly used for any kind of cell- and tissue-based assay required in biomedical applications [2]. However, researchers are well aware of the shortcomings of 2D culture, as it causes unnatural changes in the cell morphology and behavior, resulting in misleading and non-predictive data [3,4]. In contrast, 3D culture and in vitro tissue models have the advantages of providing a microenvironment to cells, which enables them to interact with the matrix and neighboring cells in more physiologically relevant conditions (Table 1) [5,6,7].

The recent advances in the tissue engineering field have fueled the imagination of scientists towards the development of in vitro 3D models, which could superiorly mimic the structural and functional aspects of the native tissues and organs [19,20,21]. There are three stimuli for this major paradigm shift in biomedical research. First is the absence of proper genetically and physiologically applicable animal model that can recapitulate human conditions [22]. Although large-scale animal models are still in use for biomedical research, they are significantly inferior in faithfully recapitulating human conditions due to substantial species genetic variation [23]. This difference also leads to poor clinical outcomes, even after successful animal trials [24]. Second is the ethical and moral dilemmas present with animal testing. Additionally, animal testing is also expensive and time-consuming and is not readily available for many researchers [25]. Finally, owing to the current advances in cell culture research [26]. Improvement in the proficiency of isolation, expansion, preservation, and guiding the growth and differentiation of human primary and stem cells towards a particular lineage has provided the means and methods for the development of cell-based in vitro 3D tissue/organ models. Applying patients’ cells for developing in vitro 3D models could not only accelerate our understanding of tissue development and genetic alteration in the disease state, but also revolutionize the disease screening, diagnosis, and treatment development. Although the superiority of three-dimensional in vitro models is not universal, it provides an edge over the 2D culture in better mimicking the structural and functional characteristics of native human tissue and organs [27].

In this regard, the goal is to utilize the combined knowledge of material science and life sciences in developing an amicable microenvironment that supports the growth and survivability of physiological features of tissue and organs on a long-term basis [28]. The essence of this supportive biological environment lies in the selection of appropriate biomaterials that can most closely resemble the physiological and structural architecture of the native matrix. From a material design perspective, among the various biomaterials, hydrogels are considered the most relevant option for mimicking the native extra-cellular matrix of tissue [29]. A significant growth in research interest of hydrogels has been seen in last three decades, partly due to the emergence of the tissue engineering field and the appealing applications of 3D culture (Figure 1). Composed of various polymeric materials (natural, synthetic, and composite), hydrogel forms an interconnected polymeric network in which it can hold a large amount of water, thus closely resembling a hydrated form of native ECM. The hydration and porosity of hydrogel also facilitate better exchange of nutrients and gases among the cells, as well as the removal of waste products. Hydrogels fabricated from natural polymers such as carbohydrates and proteins can present the biological active cues required for the growth and proliferation of cells [30,31], while hydrogel based on synthetic polymers provides tunable mechanical properties that can resemble the strength of the native tissues [32]. Since hydrogels fabricated purely from natural or synthetic polymers do not match the overall structural or functional aspects of the 3D tissues, a hybrid system was developed to overcome the shortcomings of the individual components. Hybrid hydrogels can be defined as having building blocks that are chemically, physiologically, and functionally distinctive at the microscopic or molecular level. This system might contain biologically active molecules such as protein, peptides, polysaccharides, or nano-/microstructure polymers that are connected via physical or chemical means. There are a few general strategies for fabricating hybrid hydrogels, such as in situ synthesis [33], physical blending of polymers [34], the formation of interpenetrating networks (IPNs) [35,36], and bio-conjugation [37].

Further, in addition to conventional approaches, which include cell and spheroid laden bulk hydrogel constructs and cell-cultured porous and fibrous scaffolds, numerous micro-technologies have emerged, including microspheres, microfibers, sandwich systems, micro-patterned membranes, microfluidic systems, and 3D bioprinting platforms, and they have helped facilitate the development of hydrogel-based tissue/organ models with increased fidelity. In this review article, we highlight the recent progress in the development of a 3D in vitro tissue/organ model using hydrogel as the biomaterial. We start with a broad overview of different types of hydrogel systems based on their polymeric sources, followed by a review of the various gelation techniques. A summary of hydrogel design consideration, the development of various hydrogel modules, the fabrication technologies used, and the platform developed for establishing in vitro models is also provided. Furthermore, we present 3D in vitro models of various types of tissues, including skin, liver, intestine, bone, and cartilage, as well as cancer models. Despite being such a widely used biomaterial, hydrogel presents certain challenges, which are addressed in our final discussion of various limitations and future perspectives for engineering 3D in vitro modeling. A schematic of the various components and factors in the development of composite hydrogels and hydrogel-based in vitro 3D models is provided in Figure 2.

## 2. Types of Hydrogels

Based on their composition, hydrogels have been commonly classified into natural, synthetic, and hybrid types.

### 2.1. Natural-Based Hydrogels

As these hydrogels are derived from natural polymers, they are biocompatible, interact favorably with cells, and usually show inherent bioactivity [38]. However, they have poor mechanical properties and relative instability due to their suboptimal polymer interactions and biodegradability, which affect their use in long-term cell culture [1,39]. Due to the presence of a large number of free functional groups, which include amines, carboxyl, and alcohols, hydrogels made from natural polymers can be customized synthetically to tune their properties [38,40]. Based on their chemical nature, natural hydrogels can be further classified into polysaccharide-, protein-, and peptide-based hydrogels.

#### 2.1.1. Polysaccharide-Based Hydrogels

Polysaccharides are one of the most abundant classes of biopolymers and can be derived from animals, plants, microbes, and algae. In the past few decades, researchers have focused on this class of biopolymer for hydrogel development due to its multidimensional properties, such as high stability, non-toxicity, biodegradability, cytocompatibility, availability, and its relative cheapness [41]. Although a large number of polysaccharides (dextran, guar gum, xylan, etc.) have been studied for hydrogel development, this review focuses on alginate, hyaluronic acid (HA), and chitosan-based hydrogels due to their frequent use in 3D in vitro models [42,43].

Alginate polysaccharides are widely applied as a single polymer-based hydrogel system. Alginate is composed of two uronic acids and is mainly derived from bacteria or brown algae. Since it is not a component of mammalian ECM, it lacks cell attachment motifs. However, the presence of hydroxyl side groups allows for easy chemical modifications, making it a multi-purpose material in biomedical applications [44,45]. Gelation of alginate solution is achieved by treating it with calcium chloride to form a non-covalent electrostatic complex [46]. The degree of gelation can be controlled to tune the hydrogel’s physical and chemical properties and generate more tissue mimetic viscoelastic behavior [47]. Scaglione et al. showed that by tuning the rheological behavior of alginate hydrogel, human intestine spheroids could be passaged and maintained for 90 days without significant reduction in the expression of some critical markers [48]. They also established a breast cancer model in alginate gel. However, the study also showed the need for cell adhesion motif and a more permissive environment by mixing alginate with Matrigel to demonstrate a model for invasive forms of cancer [49].

HA and chitosan are other widely used classes of polysaccharides for tissue engineering applications. Both HA and chitosan are types of glycosaminoglycan (GAG) composed of similar repeating units of disaccharides. GAGs are commonly used in combining biomaterials due to their involvement in cell signaling and communication [50]. Both polymers are biodegradable and non-toxic, albeit they have poor mechanical stability [51]. Several approaches have been used to enhance their modulus (methacrylation, hydrazide, or thiol functionalization) by exploiting the hydroxyl group of HA and the amine group of chitosan to modify their stability and degradability. In a study by Zhu et al., the stiffness of HA hydrogels was modified using bis-cystine containing matrix metalloproteinase degradable crosslinker [52]. In the same study, arginine-glycine-aspartate (RGD) peptides were used to increase the hydrogel adhesion property, after which mesenchymal stem cells (MSCs) were shown to form 3D cellular networks within seven days of culture. The same gel-based culture was used to embed noggin and BMP-2 morphogens. Under the influence of both the morphogens, the cells form a thicker and denser cellular network, with miscellaneous multi-cellular morphological features closely mimicking the morphogenesis of trabecular bone. Maji et al. used a frugal and combined method of high-speed stirring and freeze-drying to fabricate a macroporous scaffold combining chitosan with gelatin. Enhanced penetration of MSC spheroids into the construct was observed, which led to an increase in the cellular viability, migration, proliferation, and differentiation toward osteogenic lineage [53]. HA has also been extensively used in the development of cancer models for disease modeling and drug screening because it is a major component of tumor ECM. A study by Soker et al. demonstrates a successful establishment of cancer spheroid models of breast, liver, colon, and intestine using HA-based microbeads [54]. The models were demonstrated to be effective in cancer metastasis investigation, drug screening, and drug resistance evaluation.

#### 2.1.2. Protein- and Peptide-Based Hydrogels

Protein- and peptide-based hydrogels are another big class of biomaterials and are popular among materials scientists due to their biocompatibility, biodegradability, and inherent possession of cell adhesion motifs [55]. Such intrinsic bioactive attributes, including proteins such as collagen, gelatin, fibronectin, and laminin, are used in combination with other natural or synthetic polymers to increase cell attachment and growth. Most of the proteins used in hydrogel preparation are structural proteins. In the following paragraph, we provide a brief overview of a few proteins used for the preparation of hydrogels.

Collagen I is a highly abundant structural protein present in the ECM of connective tissue types found in bone, cartilage, tendon, and ligament. It is a well-established solubilized fibrillary gel used in 3D cell culture of a vast range of cell types, including mesenchymal stem cells [56], fibroblasts [57], smooth muscle cells [58], adipocytes [59], and kidney cells [60], as well as in the culture of pancreatic epithelium [61], the growth and branching of mammary epithelium [62], cancer models (skin [63], breasts [64], colon [65], etc.), and in vascularization studies [66]. The strength of collagen-based gels can be easily tuned by crosslinking them with glutaraldehyde [67], gel compression [68], solvent evaporation [69], and other chemical methods [70]. Altering the collagen gel stiffness has resulted in enhancement in endothelial cell spreading and sprouting [71]. Jabaji et al. found that a long-term culture of human intestinal organoid in collagen type I gel resulted in the formation of fully elaborated intestinal epithelial tissue [72,73]. In another recent study, Wimmer et al. used a hydrogel blend of collagen I/Matrigel for the development of human blood vessels by direct differentiation of human embryonic cells and induced pluripotent stem cells (iPSCs) in a 96-well format [74]. Some other examples of collagen I hydrogel application in 3D culture are in vitro development of murine stomach tissue and murine and human colon tissue [73].

Gelatin is a hydrolyzed derivative of triple helix structured collagen. It is water solubilized, contains RGD adhesion motifs, and is less immunogenic than collagen [75]. Although highly biologically compatible, gelatin hydrogels have poor mechanical strength. Chemical and physical crosslinking methods such as glutaraldehyde, EDC-NHS, and ultraviolet have been used to strengthen gelatin’s mechanical stability when used in in vitro cell culture [76]. Methacrylamide derivative of gelatin (GelMA) has been developed, which undergoes photopolymerization, thus enabling the tuning of the mechanical properties of the hydrogel [77]. The utilization of GelMA has tremendously increased in recent years due to it being biocompatible and non-toxic and having the potential to form vasculature networks [78]. The incorporation of nanomaterials, such as carbon-nanotube, graphene oxide (GO), and inorganic particles, has allowed for the engineering of GelMA-based hydrogels that mimic load-bearing tissues [79,80,81]. Recently, GelMA has also been applied in the bioprinting of tissue analogs [82].

Silk fibroin is another class of fiber-forming protein that has seen increasing use in tissue engineering applications [83]. Through the manipulation of some environmental factors, such as the pH, osmolarity, and shear stress, the large hydrophobic regions of the protein can be physically crosslinked to form β-sheets to form silk hydrogels [84]. Silk-based constructs have been applied to regenerate cartilage tissue by differentiating bone marrow-derived mesenchymal stem cells [85]. Self-assembling silk fibers have been used to develop minimally invasive therapy for brain-related biomedical conditions. In a study by the Kaplan group, silk/collagen-based fibrillated hydrogels were used to develop a compartmentalized structure of cortical brain tissues [86]. Skin reconstruction [87], the engineering of intestinal epithelium tissue [88], the mimicking of bone tissue [89], and nerve regeneration [90] are some of the bioengineering fields in which silk-based hydrogels and constructs have been used.

### 2.2. Hydrogels from Synthetic Sources

Synthetic polymers are usually termed inert or blank state as they do not have any biological active factors and do not promote cellular activity [45,91]. Usually, synthetic polymers are blended with natural, bioactive substances to enhance their cell compatibility [92]. Cell adhesion motifs, growth factors, or other biologically active small molecules can be covalently bonded to synthetic polymers in a controllable or anisotropic manner to guide directional cell growth and organization. The main advantage of this is that based on the various compositions, the mechanical condition (stiffness, elasticity, and durability) of the polymers can be easily defined, allowing regulation of cell growth, migration, and proliferation [93,94]. Some of the synthetic polymers commonly used in tissue engineering and in vitro cultures are polyethylene glycol (PEG), polylactic acid (PLA), poly (ε-caprolactone) (PCL), polyurethane (PU), and poly(lactic-co-glycolic acid) (PLGA) [95,96,97].

#### 2.2.1. Polyethylene Glycol (PEG)

PEG is one of the most common and favorable synthetic polymers for tissue engineering due to its chain flexibility, hydrophilicity, and low non-specific interaction with the living tissue [98]. The PEG hydroxyl end group, presents in a broad range of structures and molecular weights, can be easily functionalized (dithiol, acrylate, NHS ester, etc.). Such multi-functionality allows the incorporation of bioactive moieties through a compatible condition such as photopolymerization, Michael-type addition, or chain polymerization [99,100]. Garcia et al. showed that the modulation of mechanical properties of maleimide-terminated PEG (PEG-4MAL) incorporated with RGD motifs strongly influences the epithelial morphogenesis of kidney cells [101]. They further demonstrated the influence of mechanically tunable hydrogel in guiding tissue morphogenesis of different origins (lungs and intestines) [102].

#### 2.2.2. Other Synthetic Polymers

In comparison to natural polymers, synthetic hydrogels offer physicochemical properties that can be modulated according to the particular experimental need [103], thereby making them a preferable option in preparing ECM analogs. Examples of such synthetic polymers used in 3D modeling include polyvinyl alcohol, poly(hydroxyethyl methacrylate), poly(isopropylacrylamide) (PNIPAm), self-assembling polypeptides, and synthetic GAGs [104]. In recent years, synthetic GAG analog fabrication has been investigated due to its capability of binding growth factors, long-term availability, and control degradability. For example, in a study conducted by Chang et al., it was shown that poly(sodium-4-styrene sulfonate) (PSS) mimics synthetic heparin and has a strong binding affinity for fibroblast growth factor (bFGFs). A fabricated PSS functionalized PAm demonstrated maintenance and growth of iPSCs over 20 passages without losing its pluripotency or its morphological or karyotypic stability [105].

### 2.3. Hybrid Hydrogels

The inability of any single polymeric material to recapitulate the complexity of the in vivo microenvironment and natural ECM is well documented. The natural-based polymers contain bioactive cues for cellular growth and proliferation, but produce batch-to-batch discrepancies due to their poor mechanical and stability issues. In contrast, synthetic polymers provide excellent control over physical and chemical properties, but generally lack cell instructive motifs. Therefore, the hybridization of different compositions and types of hydrogels fabricated via various physical interactions or types of chemical polymerization provides an enticing solution. These hybrid hydrogels can be developed by blending between natural–natural, synthetic–synthetic, and natural–synthetic polymers. Natural–natural hybrid hydrogels are produced by the physical mixing of two or more different types of polymers derived from a natural source, whereas synthetic–synthetic hybrid polymers are fabricated by the copolymerization of two or more synthetic polymers to tune their biodegradability and physical properties [106,107]. Natural–synthetic hybrid hydrogels, however, are considered the most favorable for in vitro cell culture as they combine both the mechanical stability of synthetic polymers and the bioactivity of natural polymers into a single system. Further, bioactive moieties, as well as different types of nanomaterials, are integrated into the polymeric composition to impart specific properties on hybrid hydrogels [108]. In the following section, we discuss hybrid hydrogels, which we categorize based on their constituting material. A list of hybrid hydrogels along with their composition, specific features, and findings concerning in vitro cell culture applications is presented in Table 2.

#### 2.3.1. Blend of Natural and Synthetic Polymers

Hybrid hydrogels composed of polymers from both natural and synthetic sources are the most researched and reported hydrogel type, as they provide the best features of both sources [138]. The mixing of polymers, which are termed double networks or IPNs, presents the advantages of both natural polymers (cell-adhesion ligands, hydrophilicity, fibrous architecture) and synthetic polymers (controlled degradability, tunable modulus, etc.) [139]. For example, different variants of PEG, such as Poly(ethylene glycol) diacrylate (PEGDA) and poly[oligo(ethylene glycol)] methacrylate (PEGMA), have been used to bioprint hydrogels with a combination of gelatin or alginate to increase their mechanical and biological properties [2,117]. Composite bio-inks were developed by using a blend of synthetic amphiphilic peptides with keratin. The potential of this self-assembled biomaterial was studied through the development of a 3D bioprinting platform, which encapsulated cells within the modular pericellular microenvironment [116]. The term soft network composites (SNC) was recently coined based on the architecture of soft tissue, in which fibrous polymers are embedded in a weak hydrogel system. To mimic this pattern, researchers have used fibrous synthetic polymers such as PCL and embedded them in a variety of synthetic and natural polymers such as PEG, HA, and alginate hydrogels [140]. In the first and an upgrade to the above model, Bas et al. introduced the method of melt electrospinning writing (MEW) to form a continuous interconnected fibrous network of PCL combined with PEG/heparin/fibrin hydrogel [120]. This composite hydrogel system not only exhibits better pore size and mechanical properties, but also provides a suitable environment for the growth and proliferation of the human chondrocytes forming neo-cartilage in vitro. In a similar study by Visser et al., PCL microfibers were introduced into a hydrogel composed of GelMA and alginate, and a significant increase (52-fold) in hydrogel stiffness was observed [121]. The technique of SNC is promising but still at the early development stage. Further studies targeting the interaction of fibrous synthetic materials with biodegradable hydrogels and remodeling of SNC over time are in progress to develop novel matrix designs and customizations.

#### 2.3.2. Hybrid Hydrogels with Biological Factors

The ECM contains numerous biologically active macromolecules, such as cell adhesion moieties, growth factors, hormones, cytokines enzymes, and other signaling molecules, which control the spatio-temporal behavior and fate of cells [1]. PEG, PCL, PU, and other synthetic polymers as well as some natural polymers such as alginate and chitosan do not possess the biological motifs and cues of ECM. In this regard, much of the tissue engineering-based research has been focused on ornamenting biomacromolecules onto polymeric hydrogels to provide an in vivo ECM-imitating environment. One of the most common approaches is the tethering of integrin-binding peptides such as RGD and LDV in fibronectin to the synthetic polymer backbone to render the system biocompatible [141]. Further, with the advent of the third generation of biomaterials, there is a significant emphasis on the use of growth factors on biopolymers to enhance cell growth, proliferation, and in some cases differentiation or wound-healing applications. The vascular endothelial growth factor is one of the key players in the wound healing process, as it promotes angiogenesis and blood capillary formation, which are prerequisites for a successful tissue-engineered graft. In one such study, Byambaa et al. used the EDC/NHS coupling method to link the VEGF molecule with an injectable gelatin hydrogel to treat a non-union bone defect [127]. The study revealed that this hydrogel has better osteogenic and angiogenic potential, thus promoting the growth and differentiation of encapsulated cells. For certain stem cells, growth, and differentiation to a particular lineage require continuous external stimuli in the form of growth factors. In such a study, Gurkan et al. added TGF-b and FGF into the GelMA-based bio-ink [128]. They bioprinted this bio-ink with embedded mesenchymal stem cells in a growth factor gradient manner to mimic the fibro–cartilage transition in the bone–ligament interface. The study demonstrated the differentiation of MSCs into respective osteoblast and chondrocyte phenotypes in a spatially defined manner.

#### 2.3.3. Hybrid Hydrogels Incorporated with Nanomaterials

The incorporation of nanomaterials into biomaterials is a popular, versatile, and useful technique to engineer a multi-functional construct for tissue engineering applications. Nanomaterial in the form of nanoparticles, nanofibers, and nanotubes has been used to ornament hydrogels, enhancing the mechanical, electrical, and optical properties as well as augmenting the bioactivity of the system [142,143]. In this section, we briefly discuss various types of nanomaterials including metal-based, carbon-based, and polymer nanofillers, which have been used to enhance the functional aspect of biomaterials. Among many metal-based polymers, silver nanoparticles have been extensively used in combination with many synthetic polymers such as PVA and PAm to provide antibacterial properties to the construct [144]. Based on its ability to avoid the issue of particle aggregation and ensure even distribution among the gel, the in situ synthesis approach has gained popularity among researchers [145]. This technique has also been applied with many natural-based polymers such as chitosan and gelatin to obtain biocompatible and biodegradable NPS/hydrogel composites [143]. Silver nanoparticle-based hydrogel composites have been applied in skin wound repair due to their antimicrobial properties, although their high cost has prevented their popularity as an effective hydrogel system. Other metal nanoparticles such as zinc oxide, iron oxide, nickel, and silica have been used in combination with different polymeric hydrogels to impart ferromagnetic and semi-conducting properties. Iron oxide is very commonly synthesized as a ferromagnetic material and has been used in combination with many hydrogels to form ferrogels. In one study, these ferrogels were applied as actuators to mimic human muscle movement [146]. Carbon-based nanomaterials have been extensively used in the fields of engineering and medicine, with applications including multi-walled carbon nanotubes (CNT), graphene, and GO [147]. Carbon nanotubes have seen wide application in hard tissue regeneration due to their ability to increase the overall strength of the structure as well as enhance the cell attachment process. In one study, a chitosan-based polymeric material was developed through combination with CNT, which shows higher blood cell and platelet adhesion than gelatin sponges and gauges, thus increasing the ability of the material to induce a blood clot [148]. When applied to the skin wound, the material was also found to promote healing better than the commercially available Tegaderm^TM^ film. GO is another attractive nanomaterial that has been widely used to impart multifunctionality in polymer-based composite hydrogels. Tai et al. developed a GO/PAA-based material that showed excellent swelling and electrical properties [149]. In another study, Liu et al. fabricated a GO/PAm-based material using in situ polymerization, and it showed a 4.5-fold increase in tensile strength compared to PAm [150]. Polymeric nano-fillers are a class of nanomaterials that have recently become popular among researchers in the bioengineering field and include micelles, dendrimers, polymeric nanofibers, nanocrystals, and hyperbranched polymers. In one study, Maji et al. fabricated a gelatin/chitosan-based generation 4 Polyamidoamine (PAMAM) dendrimer-based 3D construct for skin regeneration [151]. The crosslinked scaffold exhibited improved mechanical strength in terms of modulus and a tensile strength comparable to that of skin, along with increased keratinocyte migration and proliferation. Nanocellulose (CNF) is a natural-based polymeric nano-filler derived from plants and bacteria and is popularly used in the biomedical and energy fields [152,153]. CNF is compatible with a wide variety of natural and synthetic polymers. Its engraftment into polymers was done to improve their physicochemical properties, particularly to create a tough and flexible biomaterial for specific biomedical applications. There are other nanomaterials, such as quantum dots [154], nanoclay [155], and calcium-phosphate nanocrystals (nano-hydroxyapatite) [156], whose incorporation into biomaterials has been found to be beneficial in terms of both physicochemical and biological attributes of the hydrogels and has been reviewed elsewhere in great detail [143,157,158].

### 2.4. Animal Matrix-Derived Hydrogels

Animal matrix-based hydrogels are usually cell-, tissue-, and organ-derived matrices and are composed of proteins (collagen, laminin, fibronectin, etc.), polysaccharides (including proteoglycans, glycoproteins, and GAGs), and other bioactive molecules (such as growth factors, cytokines, and hormones). The use of animal matrix for hydrogels for 3D cell culture has a clear advantage of better mimicking the architecture, composition, and microenvironment of native tissue. However, certain limitations such as time-consuming extraction and purification processes and batch-to-batch compositional variation have limited their popularity among researchers. In this section, we briefly discuss the application of animal matrix-based hydrogels in 3D in vitro modeling that were derived from the decellularization of tissues and organs and originated from the basement membranes of mouse sarcoma (commercially available as Matrigel).

#### 2.4.1. Decellularized ECM (dECM)

The ECM is obtained by the process of decellularization, which removes inhabiting cells from the tissue/organ, preserving only the structural and functional macromolecules and other biologically active small molecules. The process was first reported in 1948 and was later applied in 1964 for the preparation of skin homograft [159,160]. In recent years, the steady rise of patients suffering from late and end-stage organ failure and the shortage of organ donors has increased the demand for decellularized organs, which include liver [161], heart [162], lungs [163], and kidney [164]. One of the main advantages of using a decellularized matrix in organ transplantation is the ease of reestablishing vascular networks at all hierarchical levels [165]. In recent years, as an alternate to tissue- or organ-based matrices, in vitro cell-cultured-based dECMs have attracted the interest of the scientific community as they are comparatively more homogenous and can be used as in vitro models in a more controlled way, leading to results that can easily be interpreted during analysis [166]. Various types of cells, including fibroblasts, mesenchymal stromal cells, and chondrocytes, have been used to establish matrices and been investigated for their ability in cell growth, proliferation, and guiding differentiation [167]. However, it is essential to carefully consider the choice of cells for ECM production. For instance, it was observed that fibroblast-derived matrices are unsuitable for imitating the lung microenvironment in composition and complexity [168]. Nevertheless, the idea of matrix precipitation and its solubilization into hydrogels has significantly increased the utility of ECM in both in vitro as well as in vivo applications. Despite losing its architectural and structural features, the solubilized form of ECM retains all the biochemical, growth-supportive, and cell-instructive properties of its native tissue, thereby affecting the metabolic activity and morphology of cultured cells. For example, rodent and human islet cells showed an increased level of insulin secretion when cultured long-term on bladder- and pancreas-derived ECM [169]. Another study by French et al. demonstrated the use of dECM hydrogels for enhanced cardiac regeneration [170]. The porcine ventricular-derived ECM hydrogel was used to culture rodent cardiac progenitor cells, which showed enhanced cardiac marker expression compared to when cultured on collagen hydrogel. The dECM was also used to culture human cardiac progenitor cells and extended for bioprinting applications when combined with GelMA hydrogels [171,172]. These studies prompted the researchers to develop a dECM-based medical gel, VentriGel, which was first tested on patients suffering from early and late myocardial infarction [173]. Despite all the advantages, the weaker mechanical properties of dECM have limited its ability to retain an ordered structural characteristic when used as an architectural scaffold. To overcome this limitation, hybrid composites of dECM with natural and synthetic derived polymers were fabricated. Jacqueline et al. present a highly reproducible composite material containing alginate and gelatin with dECM, which was found to be mechanically stable and bioprintable and have an elastic modulus similar to tumors induced by head and neck squamous cell carcinoma [174].

#### 2.4.2. Matrigel

Despite the increased popularity of dECM in recent years, most of the in vitro 3D models in development and organotypic cultures currently use murine sarcoma-derived ECM hydrogel, which is commercially available under the trade name of Matrigel or EHS matrix [175]. Matrigel is an undefined mixture originating from the basement membrane of Engelbreth–Holm–Swarm mouse sarcoma and is composed of proteins, proteoglycans, and several growth factors [176]. Despite its well-known limitation, in the last decade, Matrigel has become a gold standard material for setting up 3D in vitro cultures and organoid development [177]. Embedding stem cells or primary cells in Matrigel under cell instructive conditions has created small but complex tissue architecture resembling the developmental stages of their organ of origin. Since the matrix is animal-derived, apart from compositional variation, it also contains the risk of carrying pathogens. ECM-based materials have indeed provided a very impressive and compelling means for cell growth and differentiation, which has enabled the scientific community to develop in vitro models that closely mimic the in vivo organ/tissue-like conditions. However, the compositional variation and complexity of the material has made it difficult to standardize protocols and to study the mechanistic biology of the targeted tissue. There is a need for a hydrogel-based biomaterial that has minimum variability, is non-immunogenic, and provides easy modification of cell instructive properties.

### 2.5. Engineered ECM (eECM) Hydrogels

In another recent approach, recombinant protein engineering was employed in the production of cell instructive hydrogels that have the advantages of both natural and synthetic polymers [178]. In this approach, the amino acid sequence of the target protein was encoded into a plasmid vector. The reengineered plasmid vector was transfected into a suitable host (e.g., *E. coli*), where the encoded genetic information was translated into the target protein. The technology allows the generation of tunable and user-defined modular protein that can integrate specific structural and functional domains of ECM. In another study, the ECM was engineered to contain an RGD motif sequence and elastin-like structural domain and used in the culture of mice intestine explant [179]. The result indicates that eECM has influence over tissue differentiation due to its adhesive and mechanical properties. The same engineered hydrogel was later used in a study to investigate the influence of the eECM’s viscoelastic and degradation properties on the preservation of stemness and differentiation ability of adult murine neural stem cells [180].

### 2.6. Electroconductive Hydrogels

It is well established that the growth and function of many cells and tissues, including cardiac tissue, neural cells, and bone cells, can be affected by electrical stimuli [181]. However, the introduction of electrical signals into general biomaterials would be difficult, as the biomaterials are barely conductive [182]. In recent years, electrically conductive hydrogels have garnered attention due to their potential in a wide variety of biomedical applications, such as the development of biosensors [183], bio-actuators [184], and drug release devices [185]. Electroconductive hydrogels (EHs) are developed by doping conductive components into conventional hydrogels. Conductive polymers such as polypyrrole (PPy), polyaniline (PANI), and poly(3,4-ethylenedioxythiophene) (PEDOT) and conductive carbon materials such as carbon nanotubes and graphene are commonly used to synthesize EHs. There are numerous studies in which EHs are used as a cell culture matrix, which cannot only provide biochemical and structural support for cells, but can also introduce more functional monitoring and manipulation of cell activities. For example, Zhang et al. prepared an electroconductive scaffold using PPy and polycaprolactone [186]. They demonstrated that the application of a 200 uA direct current to human adipose-derived mesenchymal stem cells on this scaffold promoted the migration of these cells into the inner region of the scaffold and enhanced their osteogenic differentiation. Shin et al. used an EH as a scaffold to control the contraction function of formed cardiomyocyte tissue [184]. Cardiomyocytes were cultured on a multilayer hydrogel sheet impregnated with aligned carbon nanotube microelectrodes. Upon stimulation with electrical signals, the muscle tissue was able to mimic the contraction of the heart. The study demonstrates that multilayer hydrogels work as a 3D environment for cardiomyocyte growth and that their combination with muscle tissue remarkably emulates the bioactivity of an animal heart. By responding to various stimuli such as pH and chemical signaling, EHs are also an attractive candidate for the development of biosensors. Zhang et al. developed a 3D insulated cultured scaffold from poly(dimethylsiloxane) (PDMS) with electrochemical performance by uniformly coating it with PEDOT and platinum nanoparticles [187]. The setup demonstrated desirable biocompatibility for the 3D culture of cancer cells with an excellent catalytic ability for electrochemical sensing, allowing real-time monitoring of reactive oxygen species. This in vitro 3D platform shows promise for application in anticancer drug screening and cancer treatment. Despite several emerging studies on EHs, many technical and scientific challenges remain. Further enhancement of their biocompatibility, stability, and compatibility with microelectronics would bring EHs and EHs-based in vitro devices closer to being used in clinical therapy.

## 3. Preparation of Hydrogels

Hydrogels have been fabricated by structural modifications, physical interactions, and chemical crosslinking. The triggering of various environmental factors such as temperature, pH, ionic strength, and physicochemical interaction has also been used to form physical hydrogels. Additionally, reactions such as photo-polymerization, enzymatic reactions, and other chemical crosslinking methods have been used to fabricate chemical hydrogels where the main focus is to attain enhanced stability and superior mechanical strength. Polymer surface functionalization, chemical modification, and a few other promising approaches (e.g., click reactions) for the preparation of hybrid and composite hydrogels are briefly presented in this section.

### 3.1. Physical Interaction

The physical interactions used in the fabrication of hydrogels include hydrogen bonds, electrostatic interaction, coordination bonds, hydrophobic interactions (in many instances), as well as physicochemical interactions (stereo-complexation, supramolecular chemistry, etc.) [188]. The usual approach is to form a homogenous solution by modifying ionic strength, solvent composition, and the temperature and pH. These methods are reversible and enable the solution to change to the gelation form once it reaches its initial conditions [189]. Physical hydrogels are stimuli-responsive, but they possess poor mechanical strength and often show plastic flow [190]. Physical hydrogels are also developed by crystallization [191], protein interaction [192], and amphiphilic block and graft copolymers [193].

### 3.2. Crosslinking

The crosslinking of hydrogels can be approached in three ways. The first is physical crosslinking, which includes the multiplex process of coacervation (a frequent freeze-thawing cycle that leads to the formation of cryogels) and ionic interaction. The second approach is chemical crosslinking via polymerization, co-polymerization, and covalent chemical interaction using crosslinking agents such as glutaraldehyde, borate, and glyoxal. Emulsion techniques (reverse micro-emulsion, inverse mini-emulsion), radical polymerization, photolithographic crosslinking, Schiff base crosslinking (Nucleophile addition, Michaelis–Arbuzov reaction) [194] are additional techniques used to chemically crosslink hydrogels. The final approach is grafting using irradiation, which includes gamma radiation, UV radiation, and high-energy electron beam radiation, which depends on the time and intensity of the irradiation. Crosslinked hydrogels are highly stable, with a permanently fixed shape at rest, and are usually non-reversible. However, they exhibit low extendibility and low fracture toughness. In this regard, to contain the desired advantages of both approaches of crosslinking (physical and chemical), the development of double crosslinked hybrid hydrogels was proposed [195,196]. In a study, Kondo and the research group developed a homogenous network of double crosslinked gel with tetra arm star-shaped PEG and PDMS blocks that were built together by orthogonal cross-coupling [197]. Many other double-network (DN) hydrogels have been prepared using both physical and chemical crosslinking methods, but they have the major drawback of toxicity due to the crosslinking agents [198]. Further research is required in this area to create a new generation of DN gels.

### 3.3. Chemical Modification

Chemical modification includes the preparation of complex materials with specific functions or attributes. These can include modification to provide a variety of ligands for drug delivery in sustained or burst release and the inclusion of growth factors or other biologically active molecules for imparting additional features required for targeted therapy. Lim et al. presented a protein functionalized immobilized platform by conjugating the protein with the polymer backbone of the crosslinked hybrid network [199].

### 3.4. Functionalization

Functionalization is generally carried out on the surface of the polymers/hydrogels to impart certain features to the material. Hydrogels have been functionalized by conjugating certain small molecules to reduce toxicity or incorporating features such as electrical conductivity, and enhanced strength or elasticity [194]. Additionally, functionalization helps create hydrogels with various morphologies such as hollow, multilayered, and fiber microgels [200]. Furthermore, affixing certain bioactive molecules like peptides and ligands to hydrogel surfaces adds certain types of biological functionalization and regulates cell behavior for adhesion, proliferation, differentiation, and protein synthesis, in addition to promoting specific tissue regeneration.

### 3.5. New Approaches

Recent advances in polymer chemistry have led to the design of new materials with a wide range of applications. Among such chemical transformations, “click reactions” have proved to be a valuable tool in generating materials with tunable characteristics. Reactions such as thiol-maleimide Michael addition and thiol-norbornene click reactions have attracted the attention of scientists as they are orthogonal to many naturally occurring chemical functionalities, have fewer byproducts, and the formation of intermediate thioether succinimide linkage can be modulated to give them dynamic properties [201,202]. The chemical transformation has particular utility in fabricating biocompatible hydrogels with tunable viscoelastic properties, as the developed biomaterial can carry and transfer the encapsulated cargo in an instructive way to the surrounding tissues (triggered by a change in pH or glutathione in glutathione-mediated cleavage) [203]. For example, biocompatible HA hydrogels based on copper (I)-catalyzed azide-alkene cycloaddition (CuAAC) were utilized as a drug repository tissue construct [204]. Furthermore, with the assistance of a copper-free click chemistry reaction, it is reasonably easy to reduce the toxic catalysts used in copper-catalyzed reactions. Such facilitation have been observed in radical-mediated thiol-ene/yne chemistry [205], tetrazole alkene photo-click chemistry [206], and the oxime reaction [207].

## 4. Factors Considered in the Design of a Hydrogel-Based Substrate

Cells actively interact and sense the physical and biochemical signals from their surrounding matrix and accordingly change their properties and functions, including migration, growth, ECM production, and differentiation [208]. In vitro, the cell’s functions and behavior are influenced by various substrate properties such as soluble molecules, mechanics, topography, stiffness, and degradation rate [209,210]. It should be noted that each tissue type and associated diseases differs in the types of cells present and their surrounding matrix. Therefore, the choice of biomaterial and the design of the model system are dependent on the tissue and disease type. Engineering techniques can be used to precisely control the substrate geometry, flow conditions, nutrient and oxygen supply, matrix stiffness, topography, and other local biochemical features.

### 4.1. Hydrogel Microenvironment

The in vitro development of specific 3D tissue constructs is majorly dependent on the biochemical properties of the surrounding material. Although the animal ECM-derived matrices such as dECM and Matrigel can guide cell proliferation and differentiation, their variability and poor characterization make it difficult to clearly define the cell–matrix interaction and the underlying mechanism. In this regard, chemically defined natural hydrogels are better suited for the generation of in vitro 3D tissue. For example, Broguiere et al. fabricated a fibrin-based hydrogel supplemented with laminin-111. The hydrogel was found to be conducive to the formation of both murine and human intestine epithelial 3D tissue and their long-term expansion [211]. It was thus reasoned that naturally occurring RGD motifs are crucial for maintaining stemness in intestinal cells. Additionally, defined synthetic hydrogels are used in engineering in vitro tissue models via precise control of their biochemical properties. For instance, PEG hydrogels are widely used in the formation of tissue engineering products due to their non-toxicity and ability to maintain hydration [212]. In one study, it was observed that compared to Matrigel, defined PEG-based hydrogels are better suited for inducing homogenous and polarized neuroepithelial colonies, which facilitate the formation of a dorsal–ventral patterned neural tubule-like structure [213]. The author also showed that when tuned to precise biochemical features, synthetic hydrogels can provide early neural morphogenesis.

### 4.2. Hydrogel Microarchitecture

With the help of the latest fabrication technologies such as wet-spinning, droplet microfluidics, and bioprinting, it is now possible to create a distinct internal structure of a hydrogel-based biomaterial that can provide a specific architectural environment resembling the targeted tissue. The specific topology and internal structure of hydrogels enable passive mechanical stimulation from the boundary surface of the biomaterial and allow regulation of tissue morphology, promote aggregation of cells, and promote multicellular interaction, which triggers the formation of 3D microtissues. Particular discrete structures such as microfibers and microgels fabricated by micro-technologies have been able to provide geometric confinements in the development of 3D tissues. For example, PEGDA-based microgels were formed to culture liver organoids with long-term culture and functional maturity when integrated with a perfusable chip [214]. Surface topographies have also been utilized to direct cell migration and alignment in 3D in vitro cell culture. For instance, PEG-based substrates that were formed into a specific pattern with plasma exposure and coated with Matrigel were used to generate human cardiac micro-chambers [215]. In a similar study, photopatterning was used in the development of PEG-based micro-wells, which were found to support the organization of stem cells derived from salivary glands. These micro-wells were coated with electrospun PCL nanofibers, which in turn were used in the generation of organoids with larger diameters [216]. In conclusion, hydrogel-based biomaterials can develop an in vitro cellular structure with distinct geometries and architecture by managing its topological structure, thereby guiding and improving the reproducibility of 3D tissues.

### 4.3. Hydrogel Mechanics

The mechanical properties of a biomaterial have a significant influence on the organization and development of 3D tissue. Matrices with specific mechanical properties are appealing biomaterial candidates for engineering 3D tissue formation and understanding its regulatory mechanism. Due to properties such as stiffness, elasticity, and durability, synthetic hydrogels are considered suitable analogs in the development of 3D culture. Studies have indicated that matrix stiffness is essential for regulating the integrin downstream pathway and other transcriptional activators in stem cells, which steer their organization and 3D tissue formation tendency [217,218]. In a study by Wu et al., the compressive modulus of hydrogels were prepared in lower (0.51 KPa) and upper (1.41 KPa) stiffness ranges of native brain tissue to encapsulate neural progenitor cells [219]. The result indicates increased neurite growth, proliferation, differentiation, and maturation of neural spheroids when cultured on the soft hydrogel as compared to a stiff one. In another study, PAm-based hydrogels of varying stiffness were prepared by controlling the ratio of acrylamide and cross-linkers (bis-acrylamide) to create cardiovascular organoid substrates with different mechanical conditioning [220]. The maximum differentiation efficiency and contractility percentage of cardiac organoid was obtained when the substrate had similar viscoelasticity to the native cardiac muscle (elastic modulus: 6 KPa). The phenomenon observed might be because the signal transmitted from the matrix rigidity leads to the further downstream signaling pathway in the cardiomyocytes’ proliferation and differentiation. The mechanical properties of hydrogels can be further modulated using anisotropic modification and gelation, resulting in the generation of polarized tissues.

## 5. Types of Hydrogel Units, Platforms, and Fabrication Technologies

The diverse range of functionalities exhibited by in vivo native tissues and organs are characteristics of their numerous morphological features and topographical variations. Owing to their high degree of plasticity, hydrogels can imitate the geometrical and architectural features of targeted tissues, ranging from nanometers to micrometers. Such tunable attributes can be exploited in the reconstruction and reformation of hierarchical morphologies, including the internal structure and surface topologies of tissues and organs in in vitro 3D models. Figure 3 displays various hydrogel architectural units, including micro and nanogels, microfibers, patterned membranes, and other modular frameworks that have been developed with the help of various platforms and recent fabrication technologies, which provide precise control over the distributions of cells and molecules.

### 5.1. Cell-Laden Constructs

Encapsulations of cells in 3D bulk hydrogels are one of the most simplified, approachable, and by far the most successful methods in the development of in vitro tissue constructs and clinical transplantation [221]. A broad range of polymeric materials (both natural and synthetic) such as gelatin, PEG, and alginate are available for cellular encapsulation. It is expected that the materials be biocompatible; be easily tunable to incorporate molecules that can guide cell growth, proliferation, and differentiation; and create morphological and topographical features similar to those of the real tissue. In this approach, isolated cells are typically suspended with the bulk hydrogels and crosslinked under cell-favorable conditions to mimic the in vivo conditions of the tissue. Depending on the cross-linker and crosslinking mechanism, the hydrogel could be a long-term stable construct or rapidly degradable through hydrolysis or proteolysis [222,223,224]. While stable hydrogels often restrict cell movement, and tissue deposition, degradable hydrogels allow cell migration and growth, thus promoting tissue formation to replace the hydrogel [225]. However, native tissues in in vivo conditions are far more compositionally and architecturally complex than the in vitro cell-laden hydrogel constructs. Current research trends have been focused on microtechnologies, which enable the fabrication of complex architecture through the strategic placement of cells and tissue modules.

### 5.2. Microgels

Microgels are micrometer-sized hydrogel blocs that can be used to provide geometrical and spatial confinements during the development of 3D multicellular clusters. Although microgels are generally formed in spherical geometries, they can be developed in different sizes and morphologies, including hemispheres, capsules, disks, and spindles [226,227], which can affect cell growth, proliferation, and other metabolic functions. Microgels are developed using various precursors of hydrogels, including alginate, agarose, collagen, and PEGDA [228]. A droplet-based microfluidic platform is one of the common techniques used in the development of microgels [229]. The technique has the advantage of generating and engineering the formation of individual spheroids in the form of droplets in a highly controllable manner and can be used to encapsulate single cells and multiple-cell aggregates [230]. Most droplet-based microgel formation has been developed by encapsulating cell suspension within non-adherent hydrogels such as alginate, agarose, PEG, and their derivatives, which leads to the formation of multicellular spheroids [231]. Sabhachandani et al. developed cell-laden alginate microgels as 3D tumor spheroids and studied them as a more effective preclinical drug-resistant screening model [232]. Another technique is using multiple micrometer size compartments (microwells) to trap gel-laden cells to form microgels. Such wells are usually fabricated via soft lithography technique using a PDMS stamp or mold [233]. Microwell-based spheroid formation has also been applied to investigate the paracrine effect of hepatic stellate cells [234]. Electrostatic extrusion is yet another technique for developing microgels. At the extrusion point, the jet is subjected to an electric field that generates polymer-based microparticles. Jianjun et al. used electrostatic interaction to form a porous polyelectrolyte complex (PEC) from poly(L-glutamic acid) and chitosan. Upon culture with chondrocytes, the PEC microsphere produced a significant amount of cartilaginous matrix [235]. In another study, this technique was used with decellularized adipose tissue to fabricate microgels for dynamic culture and the expansion of human adipose-derived stromal/stem cells [236]. Presently, various other techniques are used in the development of microparticles/microgels, such as the coacervation method, vibrational jet method, air suspension method, and supercritical fluid precipitation method. The details of these techniques, the microgel process formation, and their 3D cell culture applications are not covered in this article, as they have been well described elsewhere [237,238].

### 5.3. Microfibers

Under physiological conditions, several tissues exhibit tubular configuration and a fibrous structure with tissue organization-specific functions. In the past few decades, methods such as wet spinning and microfluidic spinning have excelled at controlling the compositional and geometrical features when fabricating microfibers (solid, hollow, and core-shell fibers) [239,240]. In these techniques, rapidly cross-linked hydrogels (GelMA, alginate, collagen, etc.) are preferred, as the gelation of microfibers are obtained in an aqueous solution [241,242]. Microfluidic spinning employs a specially designed microchannel that is involved in the generation of 3D coaxial flow, which consists of sample and sheath flows. Through solidification of the coaxially flowing liquid using UV light, ionic and chemically crosslinking solidified fibers can be produced [243,244]. The potential of cell encapsulation using microfluidic spinning technology has enabled its applicability in the reconstruction of in vitro complex 3D tissues mimicking organs such as the liver and pancreas [245,246,247]. For example, Kang et al. cultured L929 fibroblasts and primary hepatocytes in the center and outer layers of coaxial alginate microfibers, respectively, to prepare a liver-mimicking tissue structure [244]. Electrospinning is another popular technique used to fabricate not only micro-sized but also nano-scale fibers that closely resemble the fibrous matrix of native ECM [248,249]. The method uses an electric force to draw a charged thread of polymeric solution up to a fiber diameter of some hundreds of nanometers. The basic materials are polymers or solvent, and most natural and synthetic polymers can be electrospun into fibers. The technique uses either a single type or multiple types of hydrogels to generate a randomly packed fibrous construct (either membranous or a 3D scaffold). In one study, Honkamaki et al. developed a layered 3D construct by embedding electrospun poly(L,D-lactide) fibers in collagen hydrogel. The layer-by-layer scaffold shows better resemblance to the native microenvironment, thereby guiding the growth and neurite development of iPSC-derived neuronal cells [250]. In another recent study, the authors used a 3D hydrogel structure as a base collector instead of a common metal collector to develop a unique hydrogel-assisted electrospinning process. With various nanofiber microstructures, the process permits the forging of various types of 3D structures. This new hydrogel-assisted electrospinning method is expected to be applied to drug delivery, tissue engineering and development, and the building of anisotropic in vitro 3D tissue models [251].

### 5.4. Transwell Platforms

Transwell devices consist of permeable membranes that form a physical barrier between two different cell types, only allowing the soluble molecules to pass through. They are considered important tools in developing a 3D in vitro model that can investigate the growth and interaction of two different cell types in two separate microenvironments. Transwell platforms have been assembled with other platforms such as microchannel or microfluidic devices [252]. Commercially available transwell culture plates consist of a porous membrane suspended between two chambers. This platform has been extensively used to study drug efficacy, cell migration, and cell invasion. The majority of in vitro brain–blood barrier models have used a transwell platform in which the upper chamber contains endothelial cells and the lower chamber contains astrocytes, which represent the neural tissue [253,254,255]. Kwak et al. used a transwell platform to model 3D skin tissue. In the setup, the authors used a GelMA hydrogel to encapsulate fibroblasts and added a transwell insert to mimic the dermal tissue. HaCaT cells were cultured on top of the GelMA hydrogel to simulate the keratinocytes layer over the dermal tissue [256]. Another good model for a transwell platform is sandwich cell configurations. The sandwich culture is considered a gold standard for developing liver models, as it allows long-term maintenance of differentiated hepatocytes. In hydrogel sandwich culture, cells are first seeded onto a hydrogel substrate. The cultured cells are given sufficient time to ensure full adherence to the substrate. Thereafter, another layer of hydrogel is placed on top of the adhered cells [257]. This system of sandwiching cells between the two layers of the hydrogel can be integrated with a transwell platform and thus can regulate different cell types spatiotemporally.

### 5.5. Microfluidic-Based Platforms/Organ-on-a-Chip (OOC)

In recent years, microfluidics have become an immensely popular and sought-after platform to develop 3D in vitro models. As the name suggests, this miniaturized platform is utilized with different types of cells that can be maintained in different micro-chambers and connected by arrays of various microchannels [258]. The major benefit of this platform is the continuous perfusive flow of media, which assists in the movement of nutrients, gases, and wastes. Several studies have proved the superiority of microfluidic systems over static culture in terms of cellular metabolism and functions [259].

In the past decade, microfluidic technology-based OOC have emerged as a frontrunner in the development of in vitro 3D model systems. The OOC systems comprise various living human primary cells lined with micrometer-sized compartments and channels that can imitate the structural, mechanical, biochemical, and functional features of in vivo tissue and its microenvironment [260]. Miniaturized versions of various tissues including liver [261,262], kidney [263], heart [107], lungs [264], and intestine [265] have been developed on this platform. The simplest OOC system consists of one type of cell with a single perfused channel that can perform the function of a single tissue. For example, demonstrations of functional activity by perfusion culture of 3D hepatocyte aggregates and their responses to shear stress through the exposure of vascular endothelial cells to medium flow in a microchannel encompass some of the initial-stage research on OOC systems [266,267]. Second, this simple model has designs with two or more compartments that are connected by microchannels or porous membranes and consist of two or more different types of cells. The challenge for OOC is to create a multi-OOC design to replace the animal model experiments in preclinical studies. In 2010, Van et al. for the first time combined liver and intestines models into a single microfluidic device. Through this model, they were able to demonstrate the applicability of organ–organ interaction study on a chip, including the regulation of bile acid synthesis. Since then, many organs have been concentrated onto individual chips [268]. A study by Maschmeyer et al. described the fabrication of a microphysiological system that maintains the functionality of four organs (liver, intestine, skin, and kidney) for over 28 days on a microsystem scale [269]. Benjamin et al. fabricated a chip formed with collagen I for the study of angiogenesis and thrombosis. In this microfluidic setup, they assembled a metabolically active liver, a free contracting cardiac tissue, and a metastatic solid tumor in a biodegradable scaffold. The system successfully demonstrated a complete cancer metastasis cascade across multiple organs [270]. Studies have shown that the OOC model can be utilized in the prediction of drug efficacy and toxicity, which can provide precise estimation compared to the animal models and therefore can bridge the gap between in vitro tests and clinical trials [271]. At present, polymeric materials such as PDMS and glass are used in the preparation of OOC devices. PDMS has the benefits of optical transparency and gas permeability, which enable the OOC platform to carry out real-time and high-resolution imaging of the in vitro setup. However, it was recently observed that PDMS-based material has a major drawback in that it absorbs hydrophobic molecules from the solution and can therefore provide a misleading outcome in cell-based experiments [272]. Many hydrophilic biomaterials and hydrogels with reduced ability to interact with the molecules of the solution have been used to provide a non-absorbent PDMS surface. For example, Kuddnnaya et al. addressed this issue by employing the (3-aminopropyl)triethoxysilane (APTES)-based crosslinking strategy to stabilize the ECM protein immobilization on PDMS. The surface modification supports long term viability and adhesion of neuronal and glial cells [273]. In comparison to conventional substrate materials, natural and synthetic polymers such as GelMA, alginate, and PEG are also considered a promising substitute in the fabrication of OOC systems, as they not only have the advantage of high permeability and tunable physicochemical properties, but are also highly biocompatible and have cell adhesive properties [274,275]. Aleman et al. fabricated a GelMA-based microfluidic device with an integrated endothelial microfluidic network [274]. In another study, Zhao et al. described the fabrication of a biofunctionalized microfluidic device based on a silk protein hydrogel elastomeric material. The device exhibited well-regulated mechanical properties, long term stability in various environmental conditions, and optical transparency [275]. For the achievement of a balanced and stable structure with intricate micro-sized channels in the development of hydrogel-based OOC, different microfabrication technologies have been applied, including micro-molding [276,277], photo-patterning [278,279], and bioprinting [272,280,281,282]. With the recent progress in composite hydrogels and micro-technologies and the use of an effective combination of them, the fabrication of functional miniaturized organs to be used as disease models or in drug screening applications can soon be realized.

### 5.6. 3D Bioprinting

Conceiving the complex hierarchical 3D structure demands a highly flexible method. In recent years, 3D printing technology has gained immense popularity among researchers, in fabricating structures with tunable size, geometry, and architecture with high spatial resolution. 3D bioprinters are capable of printing multiple materials simultaneously with enhanced control over the spatial placement of various material/hydrogel-laden cells in the same construct [283]. Based on the working method, 3D bioprinting can be classified into three types: the droplet method, the micro-extrusion based method, and the laser-assisted printing method [284,285]. The advantages and drawbacks of these methods are presented in Table 3. In bioprinting, the polymers (natural or synthetic) that encapsulate cells and are used to create 3D structures are called bio-inks [286]. The physicochemical properties of the polymeric hydrogels, such as rheological properties and crosslinking mechanism, define their suitability as a bio-ink [287]. The crosslinking of bio-inks can be triggered before (pre-crosslinking), after (post-crosslinking), or during the period of extrusion (in situ crosslinking) [288]. Among all the crosslinking methods used during bioprinting, photo polymerization (light-based radical polymerization) has become the most desired method due to its flexibility in stabilizing the shape and structure of the bioprinted construct [81]. Hydrogels such as gelatin, alginate, HA, and PEG-based polymers are presently used to develop bio-inks to help create tissue-like structures [289]. Some promising studies on self- and co-assembly of materials to form composite hydrogels as novel bio-inks are also emerging [116]. It is important to note that in cell-based bio-ink printing (cells with carrier hydrogels), apart from determining printability and structural fidelity, the priority is to ensure long-term cell viability in the 3D printed construct. Numerous factors can result in low viability in the bioprinted construct, such as increased shear force during extrusion of bio-ink or harmful crosslinking methods. In one study, Lee et al. presented a versatile and novel one-step fabrication method to develop an in vitro OOC platform using 3D bioprinting. The developed platform showed spatial heterogeneity, which was used to evaluate liver tissue functionality and drug testing efficacy [290]. Other 3D printing methods such as stereolithography (SLA), digital light processing (DLP), direct ink writing (DW), laser-induced forward transfer (LIFT), and multijet modeling are suitable to print 3D hydrogel-based constructs that can subsequently be populated with cells [291]. Although the progress in bioprinting technology has been well-paced, printing tissues and organs for clinical transplantation is still far from becoming a reality. However, the method has provided a platform for creating a more sophisticated in vitro 3D tissue model that will take us one step closer to the development of real tissue and organs in a dish.

### 5.7. Organoid Systems

Organoids are 3D multicellular micro-physiological systems that are formed by proliferation, differentiation, and self-organization of primary cells or stem cells that are placed close to each other [292]. The complex and organized structure that is formed recapitulates some of the structural and functional features of the real organs [293]. Initially, this strategy was used to form cancer cell spheroids to study tumorigenesis and cell metastasis and to scrutinize cancer drugs [294]. At present, this method is frequently used for developing 3D tissue or organ reconstruction. For example, intestinal organoids have been developed from the biopsies of intestinal tissues containing intestine stem cells [295]. In this method, after selection of a cell source, a homogenous medium (the following methods are more or less similar) is used for the culture of the cells. The cells need to be cultured in free form (either suspended or embedded in a 3D-conditioned surroundings), as it enables them to expand and remodel on their own [296]. Such free growth of cells can be achieved by culturing cells in a low-attachment substrate or in a 3D microenvironment by encapsulating them in a naturally derived hydrogel that can provide the necessary instructive signals (e.g., dECM or animal-based matrix) [297,298]. The cells under suspension form clusters, proliferating and differentiating to develop into organoids (Figure 4). In the matrix-supported method, ECM components such as fibronectin and laminin provide integrin receptors to cells that maintain cell integrity and functions during the formation of organoids. Matrigel has been used widely in the development of various organoids such as cerebral organoids [299], lung bud [300], liver bud [301], gastric organoids [302], kidney organoids [303], alveolar organoids [304], intestinal organoids [305], and pancreas organoids [306]. However, due to the uncontrollable microenvironment of Matrigel, in vitro spontaneous organoid morphogenesis is not easily controlled [307]. In this regard, engineering biomaterials to obtain precise control over nutrient supply and input and output flow conditions and tunable mechanical stimulation could provide the matrix support for the growth and development of organoids [92,308,309]. Defined natural hydrogels can act as a substitution of native ECM to promote specific organoid formation. For example, Broguiere et al. developed fibrin-based hydrogels supplemented with laminin-111, which was shown to support the epithelial organoid formation and expansion [211]. Moreover, the importance of laminin was well investigated as a major biological signaling factor in the ECM during organoid formation. In addition to natural hydrogels, well-defined synthetic hydrogels have been explored in the development of organoid culture due to their tunable biochemical properties. In a recent report, Lutof et al. developed a PEG-based synthetic hydrogel platform to assist the development of embryonic stem cell-derived 3D neural tube organoids that recapitulate the key features of neural morphogenesis [213]. In another study, a laminin I-functionalized PEG hydrogel system was shown to promote the formation and expansion of organoids from pancreatic progenitor cells [310]. To date, several organoid formations have been explored, including skin [311], pancreas [312], lungs [313], liver [314], kidney [315,316], and brain [299,317]. Even though these organoids lack complete features of real organs or tissue, their cellular assembly and heterogeneity have made them a suitable platform for screening drugs and developing disease models as an alternative to the present 2D cell-based assay and animal models [318].

## 6. Engineered 3D In Vitro Models

In this section, we focus our discussion on various studies on the development of 3D models of specific tissues/organs in the past decade that have used hydrogels as biomaterials. A selected survey of recent in vitro models of various tissue types is provided in Table 4.

### 6.1. Skin

The skin is the largest organ of the human body, covering an area of ~2 m^2^, and is the body’s primary barrier against most environmental agents, including pathological microbes and chemicals. Culturing of human 3D skin ex vivo biopsies began early, in the middle of the twentieth century [319]. However, 2D culture later prevailed as standard lab practice, as it was much simpler and more reproducible than organ/ex vivo culture. Recent development in cell culture techniques and 3D fabrication technologies has seen an increased momentum toward research and development of in vitro 3D skin models, as indicated in numerous studies [320]. Skin equivalents containing the epidermis and dermis layers developed from keratinocytes and fibroblasts were the first type of 3D in vitro skin model [321,322]. Among several, collagen-blended hydrogels have emerged as the most worked on substrate when producing skin equivalents [323]. In one study, a collagen-based hydrogel containing dermal fibroblasts was deposited followed by another layer of keratinocytes and melanocytes to mimic skin tissue [324]. Stroebel et al. exhibited the development of scaffold-free spherical skin microtissues containing different layers of keratinocytes and a dermal fibroblast core [325]. Scaffold-free models are considered more advantageous due to their simplicity, low cost, easy reproducibility, and suitability for high-throughput biochemical analysis. However, they lack proper skin architecture and do not accurately recapitulate the liquid–air interface of physiological skin. In this regard, composite hydrogels are gaining popularity in skin tissue engineering, as the skin matrix is composed of several ECM components, such as collagen, fibronectin, elastin, vitronectin, and GAGs. In a recent report, a hydrogel produced from blended collagen and silk was presented as an ideal dermal material [326]. The silk helps stabilize the structure, whereas the combination of collagen presents the cell-binding domain to the construct. This blend proved more resistant to temporal degradation than one made from only collagen. In another study, using a mixture of gelatin, alginate, and fibrin, Pourchet and colleagues were able to bioprint a full-thickness skin model with a stratified epidermis [327]. Although extensive work has been done in 3D skin models, the key limiting factors in such skin equivalent models are vasculogenesis and functionally crucial appendages such as sebaceous glands [323,328]. Skin organoid models developed from iPSCs have been shown to develop sebaceous glands along with hair follicles in mice over 30 days [311]. Finally, issues such as the incorporation of immune systems and vasculature can be addressed with the current development and convergence of different fabrication techniques.

### 6.2. Bone

The musculoskeletal system determines the body’s shape and is important for locomotion in vertebrates. Bone is one of the key components of the musculoskeletal system and plays a vital role in providing support, protecting organs, distributing force, and producing blood [329]. Bone is a natural composite and presents a unique hierarchical structural organization at multiple scales that grants it the required toughness [330]. To mimic the in vivo conditions of bone, researchers focused on the 3D spheroid construct. Gurumurthy et al. were able to show enhanced osteogenic functionality upon differentiation of MSC spheroids when compared to the conventional 2D culture [331]. To imitate the in vivo conditions more adequately, methodologies such as culturing of osteoblasts and endothelial cell types together to develop vascularized bone tissue were utilized. When HUVECs and osteoblast cells were co-cultured in a collagen matrix to form a spheroidal model, both osteogenic differentiation and vasculogenic tube-like structures were found [332]. It is well known that bone is a highly vascularized organ and engineering a vascularized and mineralized matrix requires synergistic interaction between osteogenic and endothelial precursors [333]. At present, vascularization strategies include using (i) angiogenic factors in combination with 3D scaffolds, (ii) pre-vascularization methods, and (iii) co-culture systems to engineer vascularized tissue constructs (Figure 5) [334,335]. For example, the Braghirolli group showed that the PCL scaffold constructed by electrospinning and loaded with VEGF molecules encouraged the penetration and growth of endothelial cells within the 3D matrix [336]. The co-culture method is comparatively more complex, as several parameters are considered for a successful vascularization outcome, including cell types, cell seeding methodology, 3D construct, media, and microenvironment. In recent work, Tsigkou and colleagues presented simultaneous osteogenic differentiation and vasculature development by combining bone marrow-derived MSCs and HUVECs in a 3D scaffold and hydrogel, respectively [337]. The result demonstrated capillary structure formation with three to seven days of culture when implanted in a mouse model. The hybrid scaffold has shown potential in the field of bone tissue engineering. Dhivya et al. constructed a hydrogel containing zinc-doped chitosan-HA-β-glycerophosphate and demonstrated the material bone formation ability both in vitro and in vivo without any toxic effect on cells [338]. In another instance, Zhai et al. constructed a bone cell-laden hydrogel consisting of PEGDA, HA, and nanoclay [339]. The nanocomposite showed enhanced osteoconductive properties in long-term culture, rationalized to be due to the presence of the bioactive ions of nanoclay. The use of 3D bioprinting using a composite hydrogel of alginate, gelatin, and hydroxyapatite as bio-ink has also been demonstrated to support mesenchymal stem cells and differentiation toward osteogenic lineage [340]. Among the various bioprinting techniques, extrusion-based bone printing is the most common, as it uses the hydrogel as bio-ink with varying viscosities and high cell densities. Levato and colleagues constructed a composite biomaterial consisting of PLA microcarriers with GelMA/gellan gum [341]. They used bio-ink combined with a high concentration of MSC microtissues to fabricate a bone construct via bioprinting. The in vitro studies demonstrated that the strategy allowed higher cell adhesion, proliferation, and differentiation with enhanced bone matrix deposition within the construct. However, due to the high mechanical properties of bone, hydrogels are not suitable for fabricating larger voids or hollow spaces within the construct, as they will collapse the structural features. In this regard, a sacrificial material, such as poloxamer F-127, can be introduced for printing constructs with voids for enhanced perfusion and subsequent vascularization [342,343].

### 6.3. Cartilage

Cartilage is smooth elastic tissue providing padding and protecting the end of the bone [344]. It consists of sparsely spread specialized cells called chondrocytes embedded in a large amount of ECM. Due to the avascular nature of cartilage, its repair and regeneration are slow and difficult [345]. With the increasing cases of arthritis and clinical demands, cartilage regeneration has gained much attention in the tissue engineering field. Several hydrogel-based methods have been used to develop chondrocyte niches and realistic microenvironments [342,346]. Within the last decade, lithography and 3D bioprinting have become a popular approach for this purpose. In a stereolithographic bioprinting approach, Lam et al. used methacrylated HA and methacrylated gelatin as bio-ink to create a cartilage model with varying chondrocyte densities. The model maintained its shape, cell distribution, and viability for over 14 days and demonstrated cartilage proteoglycans and type II collagen deposition. By increasing the densities of cells, the model showed a higher differentiation pattern that led to enhanced cartilage-typical zonal segmentation [347]. In another instance, HA/polyurethane was used as hydrogel material to print a 3D MSC-laden construct for cartilage formation [348]. The results showed that MSCs differentiate successfully into chondrocytes, whereas the matrix demonstrates mechanical properties similar to those of cartilage. Moreover, as cartilage is avascular and aneural, most of the stimuli received by chondrocytes are mechanical. In this regard, Paggi et al. developed a versatile microfluidic platform that supports evaluating the impact of various mechanical stimuli on chondrocytes up to single-cell resolution. This platform will be instrumental in studying the progression of various diseases and answering biological questions [349].

### 6.4. Liver

The liver is considered an important organ in detoxifying chemicals and metabolizing drugs in the body. The development of a 3D in vitro model of the liver can not only help to unravel the physiological phenomena in the liver, but also provide a platform to accurately predict drug effects and toxic responses [269]. Despite being capable of rapid regeneration, the liver remains in high demand for organ transplantation due to liver tissue damage from multiple factors [312]. From a material perspective, due to the low mechanical strength of the liver, hydrogel-based composites are an ideal candidate in engineering in vitro models. In one study, Ma et al. produced mechanically flexible composite hydrogels from GelMA to support iPSC-derived hepatic cells and glacial-methacrylate-HA to enhance endothelial cell growth and proliferation [350]. The research group used the above composite material as bio-ink to successfully bioprint a patient-specific hepatic model that closely mimics the architecture and cell functionality of native tissue. Mazzocchi et al. used a composite hydrogel of HA and collagen to print hepatic tissue that was shown to preserve the native tissue microenvironment [351]. The tissue construct used hepatocytes and stellate cells as cell sources and was found to be responsive against the effects of common liver toxicants. Although the 3D liver models can sustain in vivo-like conditions for several days or weeks, the static culture situations do not allow the movement of accumulated medium, which can have a toxic or self-inhibition effect on cell viability and functionality (urea or ammonia accumulation). In this regard, a microfluidic platform or liver-on-a-chip could be utilized to recapitulate the in vivo flow rate in removing the metabolites and other functional products. In a study by Rennert et al., a two-channel microfluidic platform was created to develop an in vitro 3D liver model [352]. In one of the channels, the vascular layer was developed composed of endothelial cells and macrophages, whereas the hepatic model was composed of HepaRG cells co-cultured with stellate cells. Both models were separated by a membrane mimicking the space of Disse. The results showed hepatocyte polarity and the allowance of hepatobiliary function. In a later follow-up study, an oxygen measurement system was incorporated for toxicological screening [353]. In another study conducted by Lee et al., porcine liver dECM was used as bio-ink to print a 3D liver-on-a-chip platform. Apart from hepatocytes, the study also included endothelial cells and cholangiocytes to mimic the vascular and biliary systems in the platform. The creation of a biliary system enhanced the liver-specific function and increased the drug sensitivity response many fold when compared to a chip without a biliary system [354]. In a later study, the same group developed a liver fibrosis-on-a-chip platform using dECM and gelatin bio-ink and activated stellate cells. The platform exhibited increased collagen accumulation, cell apoptosis, and reduced liver-specific functions, all of which are characteristic features of liver fibrosis [355]. In a multi-organ-level study, an OOC was developed to recreate the metabolic dynamics connecting gut epithelial cells to liver cells [356]. In such a multi-organ-level system, it is possible to study the gut–liver axis, which could also include immune cells, to study inflammatory responses, investigate diabetes, and develop fatty liver disease models [357].

### 6.5. Gastrointestinal Tract

In recent years, there has been a significantly growing interest in 3D in vitro model development of intestine tissue due to the increasing demand for food science and toxicology analysis and disease understanding. The intestinal epithelium is a multitasking tissue containing different cell types specialized in different functions: enteroendocrine cells, goblet cells, Paneth cells, and microfold cells [358]. Roughly 90% of the food digestion and absorption in the digestive tract happens in the small intestine [359]. The intestinal epithelium consists of highly polarized tissue with a defined 3D tissue architecture (Crypt-villus organization). Several materials and microfabrication techniques have been used in the generation of a 3D model for mimicking the topography of the gastrointestinal tract and to study its impact on cellular behavior. One of the first studies in this regard used a combination of molding techniques to generate a scaffold that represents intestinal villi in a collagen hydrogel [360]. Morphological similarities with human villi were found when Caco-2 cells were cultured on the construct for three weeks. The relevance of this system was further assessed by studying drug permeability and the role of mucin (MUC17) in antibacterial response [361,362]. Gjorevski et al. designed a composite hydrogel composed of PEG/laminin with RGD motifs [92]. The designed materials were initially optimized for stem cell proliferation, but were later observed to allow cell differentiation and organoid formation, presenting an alternative to Matrigel in intestinal organoid development. Kim et al. employed 3D bioprinting using collagen-based bio-ink to form a mesh structure with a crypt compartment and vertical protrusion to mimic the villi architecture [363]. The model was developed with external Caco-2 cells forming the epithelium and internal HUVECs to reproduce the capillary structure, and the results exhibited an increased proliferation rate and expression of differentiation markers. In another study, high-resolution stereolithography 3D printing was used in the formation of a PEG-DA-based hydrogel to form a 3D structure containing both crypts and villi [364]. The cultured Caco-2 cells exhibited increased polarization and expression of enterocyte differentiation markers, which also demonstrated a strong influence on the cell behavior of the hydrogel material and topology of the construct.

### 6.6. Cancer Model

Despite significant accomplishments in the field of biomedical treatments and therapy, cancer is still the leading cause of death worldwide. For the past four decades, the conventional 2D model was used to study tumor progression. However, it does not properly mimic the heterogeneous architecture of the native tumor microenvironment [365]. Therefore, there is a need to develop a 3D tumor model that can present all the necessary physiological characteristics [366]. The first 3D in vitro culture to be efficiently demonstrated was a cancer spheroid culture [367,368]. In recent years, with the help of 3D bioprinting techniques, in vitro cancer models have been developed that can mimic the 3D complexity of native tissue. Such models are used in the study of cancer pathophysiology as well as for the screening of anticancer drugs [369]. In this regard, numerous polymeric composite materials have been featured as bio-inks. In a study reported by Zhao et al., gelatin-alginate-fibrinogen composite bio-inks laden with cervical cancer (HeLa) cells were printed to form a 3D porous structure that stabilized by crosslinking with calcium chloride [370]. The sensitivity of the chemotherapeutic drug paclitaxel was evaluated, and it was found to be more effective on HeLa cells on 3D structures compared to 2D cultures. A study conducted by Beck and colleagues used a composite hydrogel of PEG/Matrigel to investigate cancer cell metastasis [371]. The matrix rigidity was controlled by increasing the degree of crosslinking of the PEG polymer, whereas the cellular adhesion signals were presented as peptide-conjugated cyclodextrin incorporated into the PEG networks. The study revealed that nominal values of cell adhesion and rigidity of the PEG matrix induce the migration of mammary malignant epithelial cells. Given the complexity and variability of tumor niches, researchers are striving to develop patient-specific tumor models that can be used in understanding tumor progression, diagnosis, and treatment [366,372]. The work performed to date has demonstrated the flexibility of hydrogels in investigating cancer pathophysiology. Further research is needed in the direction of the generation of matrices and their interaction with cancer cells to develop a better model mimicking in vitro tumors.

**Table 4 ijms-23-02662-t004:** Selected survey on hydrogel types and fabrication technology used in in vitro models of various tissue types.

Hydrogel Types	Fabrication Technology	Features	References
Tissue Type: Skin			
Collagen, fibrinogen, and sodium hyaluronate	Bioprinting	A layer of thrombin was co-delivered to induce gelation of fibrinogen. The printed sheet could form in situ on murine and porcine wound skin with any topography.	[373]
Silk and collagen		Due to the presence of the cell-binding domain of collagen and stabilizing properties of silk, the scaffold exhibited more resistance to time-dependent degradation.	[326]
Gelatin, alginate, and fibrin	Bioprinting	A full-thickness skin model with a stratified epidermis was developed.	[327]
GelMA	Photocrosslinking	A skin model with an epidermal-like structure combined with an air–liquid interface was developed.	[374]
Silk and PCL	Electrospinning	The scaffold possessed a surface topography that promoted fibroblast-induced collagen deposition.	[375]
Tissue Type: Bone			
Gelatin, chitosan, and hydroxyapatite	Freeze drying	The macroporous architecture allowed greater migration of MSC spheroids and led to a greater degree of mineralization of the construct.	[53]
PLA and collagen	3D printing	The porous disc-like construct was shown to support the growth and proliferation of osteoblasts, fibroblasts, and endothelial cells and induce neo-vessel formation.	[376]
Cellulose/BMP-2	Electrospinning	Bone marrow-derived MSCs showed oriented growth aligned with the underlying nanofiber morphology as well as increased alkaline phosphatase activity and calcium deposition with rapid rabbit calvaria wound repair.	[377]
Poly polystyrene sulfonate and collagen I	Ice templating	The interaction of human adipose-derived stem cells with electroactive 3D scaffolds was analyzed. The results highlighted the usefulness of porous conductive scaffolds as 3D in vitro platforms for bone tissue models.	[378]
PEG-DA	Stereolithography	An osteogenesis-on-a-chip device was developed that supports the proliferation, differentiation, and ECM production of human embryonic stem cell-derived mesenchymal progenitor cells for an extended period of 21 days.	[379]
Tissue Type: Cartilage			
HA	Electrospinning	A 3D nanofibrous scaffold was developed with crosslinked HA. The results showed a superabsorbent property, elastic behavior, and good cytocompatibility.	[380]
GelMA and methacrylated HA	Stereolithographic bioprinting	The 3D-printed model maintained chondrocyte distribution, differentiation, and ECM formation. Both materials showed cell viability and phenotype maintenance for a period of 21 days.	[347]
GelMA and tricalcium phosphate (TCP)	Co-axial extrusion bioprinting	The osteochondral defect was reconstructed by developing an in vitro 3D calcified cartilage tissue model. An investigation of a gene expression study confirmed the effects induced by ceramic nanoparticles in the differentiation of MSCs toward hypertrophic chondrocytes.	[381]
Agarose	Soft lithography	A versatile platform of articular cartilage-on-a-chip that can provide 3D multi-axial mechanical stimulation on a chondrocyte-loaded hydrogel was developed.	[349]
Methacrylated gelatin	Lithography using silicone mold	Microphysiological osteochondral tissue chips derived from human iPSCs were developed to model the pathologies of osteoarthritis (OA). Celecoxib, an OA drug, was shown to downregulate the proinflammatory cytokines of the OA model.	[382]
Tissue Type: Liver			
Agarose	Self-aggregation of iPSCs	In vitro 3D liver tissue that exhibited a stable phenotype for over one year in culture was generated. The study presented an attractive resource for long-term liver in vitro studies	[383]
Agarose-chitosan	Liquid-cryo bath treatment of polymeric molds	At neutral pH, the negative charge of the scaffold surface ensured cell–cell interfacial interaction, followed by colonization of hepatocytes. The in vitro studies also indicated enhanced cellular metabolic activity.	[384]
Alginate and Pluronic F-127	3D bioprinting	iPSC-derived hepatocyte spheroids recapitulated liver epithelial parenchyma using 3D bioprinting.	[385]
Basement membrane extract	Soft lithography	A sinusoid-on-a-chip was established using four different types of liver cells (hepatocytes, endothelial cells, stellate cells, Kupffer cells). The study was the first to report the application of a liver chip in assessing the effect of hepatoprotective drugs.	[386]
Gelatin and porcine dECM	3D bioprinting	The study developed a liver fibrosis-on-a-chip platform using dECM and gelatin bio-ink and activated stellate cells. The platform exhibited increased collagen accumulation, cell apoptosis, and reduced liver-specific functions, which are characteristic features of liver fibrosis.	[355]
Tissue Type: Gastrointestinal tract			
Silk	Freeze drying	Tissue characterization showed four differentiated epithelium cell types (enterocytes, goblet cells, Paneth cells, enteroendocrine cells) along with tight junction formation, microvilli polarization, low oxygen tension, and digestive enzyme secretion in the lumen.	[387]
Thermo-responsive Novogel	3D bioprinting	The histological characterization of intestinal tissue demonstrated an injury response against the compound-induced toxicity and inflammation.	[388]
Collagen IV and Matrigel	Soft lithography	A human duodenum intestine-chip was developed. The in vitro tissue presented a polarized cell architecture with the presence of specialized cell subpopulations. It also demonstrated relevant expression and localization of major intestinal drug transporters.	[389]
Colon-derived dECM	3D bioprinting	The bioprinted intestinal tissue models showed spontaneous 3D morphogenesis of the human intestinal epithelium without any external stimuli.	[390]
Poly(3,4-ethylenedioxythiophene) doped with poly(styrene sulfonate) (PEDOT:PSS)	Freeze drying	A tubular electroactive scaffold served as a template for a 3D human intestine and enabled dynamic electrical monitoring of tissue formation over 1 month.	[391]
Tissue type: Cancer			
Poly-l-lactic acid	Thermally induced phase separation	The study generated scaffolds with different morphologies, porosities, and pore architectures and indicated that a pore size ranging from 40 to 50 μm induces tumor cell aggregation and the formation of the irregular tumor masses typically observed in vivo.	[392]
Matrigel	Organoid formation	A lung cancer organoid from patient tissue was established, the tissue architecture was recapitulated, and the genomic alterations of the original tumors were maintained during long-term expansion in vitro. The model responded to cancer drugs based on their genomic alterations and could be useful for predicting patient-specific drug responses.	[393]
Fibrinogen and Matrigel	Photolithography	A platform that imitates the mass transport near the arterial end of a tumor microenvironment was fabricated. An observation of the hallmark features of tumor progression was provided.	[394]
Matrigel	Photolithography	A colorectal tumor-on-a-chip model was developed. The platform can validate the efficacy of drug-loaded nanoparticles.	[395]
GelMA, alginate, and PEG-DA	Bioprinting	A tumor model that includes a hollow blood vessel and a lymphatic vessel was fabricated. The ability of imitating the transport mechanisms of drugs inside the tumor microenvironment was demonstrated.	[396]

## 7. Evaluation of In Vitro Models

The current biotech and pharmaceutical industry experiences high research and development costs and overall low success rates in the screening and development of new chemical and biological entities as therapeutic agents. Between 2005 and 2014, the primary reason for termination was poorly validated targets with little human relevance and lack of human efficacy at phase II/III (35%) [24]. There is a growing need for the development of human-based models that can be applied to healthy or disease states at the early stages of drug delivery. In recent years, numerous 3D models have been investigated and developed by a number of academic institutions and industries due to the emergence of innovative micro-technologies (spheroids, organoids, artificial scaffolds, lithography, and bioprinting). However, there is a significant gap between the development and qualification of such in vitro models, without which the end user cannot be confident enough about the physiological relevance of the system and subsequently the data produced from the model.

The standard rationale for any in vitro model is to streamline the experimental variables and fundamentally segregate different modules of organs or organ-like structures to study under well-organized and easily assessed conditions. How precisely these conditions recapitulate the in vivo conditions depends upon the study design and outcomes. A model should be qualified by deciding its domain of validity as defined by Scanell and Bosley based on rigorous scientific data and not on assumptions such as “primary cells are better than immortalized cell lines” or “a more complex model is more relevant” [397]. Different in vitro models represent different levels of cellular organization and behavior, and one classified as right must be fully qualified with regards to its qualities in contrast with the in vivo situation.

Many in vitro 3D models have been introduced into the literature with characterizations that have relied on techniques looking at a limited number of biological and physical descriptions, such as histological assessments, measurements of gene and protein expression at the subpopulation level by reverse transcriptase-polymerase chain reaction (RT-PCR), and enzyme-linked immunosorbent assay (ELISA)-based assays. More in-depth characterizations of models using the advances in omics technologies, including metabolomics, transcriptomics, and proteomics at the single-cell level, will enable a comprehensive understanding to score a model and assess whether it is fit for purpose. For example, a recent study by Kasendra et al. presented transcriptomic profiling, which showed there was an increased similarity in gene expression profiles when comparing a microfluidic intestinal organoid to adult duodenal tissue than to static duodenal organoids [389]. The study demonstrated a model that better replicates the in vivo tissue and thus increases the domain of validity for the model for possible drug transport metabolism and toxicity studies. The in vitro model also showed the presence of polarized cells, intestinal barrier function, the presence of specific cell populations, and in vivo–like expression, localization, and function of key intestinal drug transporters, which are necessary requirements for intestinal drug metabolism cellular models.

One of the important variables that determines the key aspect of developing, qualifying, and implementing in vitro models is cell type (cell lines, primary cells, stem cells). Immortalized cell lines have a distinct advantage, as they can be obtained in large numbers, which makes them an attractive candidate for in vitro model development for target validation work. With the aim of determining the relevance of the cells for use in research, the Open Targets project (in collaboration with Wellcome Trust Sanger Institute) has taken a number of cell lines used in research and performed a detailed investigation of gene expression with reference to primary cell data [398]. This study will provide a useful resource for the assessment of cell lines and physiological relevance with respect to different tissue types for use in in vitro models. Stem cells, on the other hand, have a number of unique advantages, as they retain the genetic information of the donor, but are scalable and amenable to gene editing and represent an infinite source of cells for target validation in in vitro models. Another benefit of stem cells from originator samples is the generation of organoid cultures, which is considered to mimic more closely the development and niche observed in vivo.

Development of a closely mimicking microenvironment in the model is another important criterion to evaluate. Many hundreds of studies to date have been published in which polymers from synthetic and natural sources and their blends have been used to create 3D constructs in many physical forms (e.g., gels, fibers, weaves, meshes, sponges, foams, channels), asserting their relevance based on cell adhesion, viability, proliferation, and superiority over 2D culture methods. Although comparisons of 3D models with traditional culture methods have merit, they alone cannot justify the significance of improved models. Similarly, many attempts have been made to seek a “one-size-fits-all” matrix, accommodating any cell type and many culture platforms, instead of developing organ-, tissue-, or pathology-specific models that can support clinically relevant pathways. Under normal physiological conditions, an ECM derived from different tissues exhibits its own unique architectures, mechanical properties (e.g., elastic energies, moduli), protein compositions, and molecular complexities [399]. These properties can be modulated as a function of normal processes, such as wound healing, or via pathological manifestations, such as cancer induction [400]. Instead of pursuing the development of a universal ECM mix, more focus should be placed on producing easily modified matrix components that can be mixed and matched to closely re-create native biochemistry and matrix rigidity. Another significant challenge in the development and implementation of in vitro models is the limitations imposed by static cellular models. Incorporating fluidics into models to mimic blood supply and interstitial flow is likely to increase the physiological relevance of the model [401]. The current microfluidics-based OOC models have solved this problem to a great extent [23]. Technological advances such as bioprinting have reproducibly demonstrated the ability to spatially control the deposition of multiple cell types and gels, resulting in the construction of tissues with architectures closer to those of organs [402]. Advances in big data and the capabilities of artificial intelligence (AI)-based largescale computing methods such as machine-learning (ML) and deep learning (DL) have the capacity to enable the multiparametric optimization of future in vitro models as well as to qualify and validate the clinical translatability of the models. Hepatotoxicity or drug-induced liver injury (DILI) is one of the most prominent areas in which such data-driven methods are used to support, optimize, and cross-validate the emerging complex in vitro models. Another good example is an ML-based platform, CANscript, developed by Mitra Biotech, where the data from in vitro models were used to train an ML-based algorithm to give a predictive translatability score representing the clinical efficacy of the drug response [403]. This shows the potential of computational and data-driven methods for building a more human-relevant in vitro model.

## 8. Challenges and Future Perspectives

Three-dimensional in vitro models are much more relevant compared to 2D culture in terms of imitating the complex physiological and pathological processes of native tissue and organs. These 3D in vitro platforms can potentially be used (i) to screen drugs and molecules for their safety and efficacy, (ii) to study the physiological processes in basic biology, and (iii) to facilitate clinical transplantation. As discussed in the previous section, several in vitro models targeting different tissues have been developed using various types and compositions of hydrogels. At this juncture, it is fair and appropriate to say that hydrogels are one of the basic requirements and logical choices in supporting 3D in vitro tissue model development. The simple reason behind this is they can inherently mimic the native tissue structure, which contains a complex polymeric network in a high-water-content environment. Hydrogels have been used in multiple 3D in vitro models to mimic the extracellular environment of the tissue. However, multiple factors need to be considered for composite hydrogels to recreate a 3D microenvironment of native tissue. In this section, we discuss the current challenges and future directions concerning hydrogel-based material design, recreation of the dynamic tissue microenvironment, and multi-tissue connectivity during the engineering of 3D in vitro models.
(i).The native ECM contains biochemical cues such as adhesion ligands, growth factors that are not evenly distributed throughout the matrix. Such anisotropic features in vivo are important in guiding cellular behavior and fate. Most of the hydrogel systems currently exhibit isotropic properties and completely lack anisotropy of the tissue microenvironment, which does not allow design flexibility in controlling hydrogel properties dynamically. In future studies, patterned systems could be developed containing gradient features over a hydrogel backbone for directing cell behavior. In this regard, stimuli-responsive smart hydrogels have attracted much attention, as they allow dynamic changes in their properties in the response under the defined stimulus. Novel hydrogel-based ECM mimetic formulations must be studied and investigated to contain features of anisotropy as well as flexibility (in stiffness, degradation, topography, etc.).(ii).Tissue formation, disease development, and post-disease progression are all dynamic processes. Therefore, in vitro models must recapitulate such dynamic features of tissues and spatiotemporally control the features of the matrix, including presentation of biochemical features and topological and viscoelastic properties. In this regard, a reversible crosslinking strategy could be incorporated into the composite hydrogel system to control the degradation and mechanical properties of the synthetic matrix for supporting cell activity and long-term culture.(iii).The major limitation of the present 3D in vitro model is its inability to completely mimic the complex features of tissue and disease microenvironments. The hierarchical design of organs, ranging from macroscale to micro and nanoscale, contains a complex structural arrangement of tissues that further contain a wide range of cell types with numerous cellular compositions and organizations. It is a challenge to design a tissue microenvironment in high resolution (concerning scale) that exhibits such variabilities and versatility of tissue arrangements. With the present technological advancement, it is quite difficult to exactly mimic a native tissue microenvironment. An ordered approach could be developed to determine the complexity needed for the 3D culture. In the long term, the development of an extensive database of various types of hydrogels, their interaction with cells, and other available knowledge of their features can be integrated with big data analytics and AI to predict definite factors in the development of a targeted model. Such an interdisciplinary approach would help us not only to understand the complexity of tissue niches, but also to develop and design composite hydrogels that would guide cell fate in the relevant microenvironment.(iv).The last obstacle would be to integrate multiple tissue types or diseases in a single closed-loop platform. It is well understood that organs do not work in isolation, but are always working together and communicating with each other via biochemical cues, thus affecting and controlling each other directly or indirectly. For example, any toxic drug that affects the liver also influences the functionality of heart and lung tissue [281]. Similarly, in cancer, to metastasize the malignant cells, different tissue models are required, which can be connected via circulatory channels [404]. In this direction, multi-organ platforms have already been developed with some success [281]. However, the development of 3D systems with the above-mentioned features is highly desirable in establishing a functional and predictive model.


Although a 3D cell culture system provides a more physiologically relevant microenvironment as compared to 2D tissue, it can also be a source of shortcomings with respect to experimental analysis and detection. As most of the current methods of measurement are based on 2D cell culture, they might not be compatible with the 3D cell culture system. For instance, to determine cell number or quantify cell viability, most of the current methods rely on fluorescence-based analysis or manual counting of cells after trypsinization. These methods have a major limitation when applied in a 3D cell culture system. Therefore, novel methods and protocols that can provide accurate analysis in a 3D cell culture system must also be established.

## 9. Conclusions

Hydrogels are a class of biomaterial that has been developed to mimic the composition and structure of the native tissue matrix in a 3D in vitro tissue model. A range of biopolymers is used to fabricate hydrogels that can support and direct cell behavior and function in the desired way to generate 3D tissue-like culture. Due to the limitation of a single type of polymer, hybrid hydrogels are proposed with a combination of natural and synthetic polymers along with the amalgamation of a wide range of nanomaterials and biological factors that can impart specific tissue niche-like features. The development of dECM-derived hydrogels has presented a good alternative and opened up a new way to engineer the physiology of and simulate 3D in vitro models. This review article has summarized important advancements in the field of hydrogel biomaterial used in the generation of 3D tissues and found that hydrogels are becoming an attractive choice in emulating the 3D native matrices toward in vitro model development. Although research regarding hydrogels is still in its infancy, and many more technical challenges remain to be overcome, recognition of the reliability and reproducibility of hydrogels for specific in vitro models will be required for their acceptance in the therapeutic and pharmaceutical industries.

## Figures and Tables

**Figure 1 ijms-23-02662-f001:**
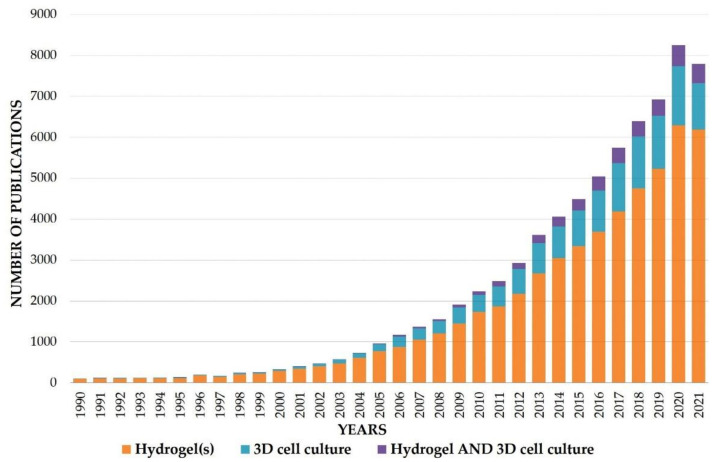
Number of publications related to “hydrogels” and “3D cell culture” from 1990 onward. The total number of studies was calculated based on data from PubMed (www.pubmed.ncbi.nlm.nih.gov (accessed on 20 November 2021)).

**Figure 2 ijms-23-02662-f002:**
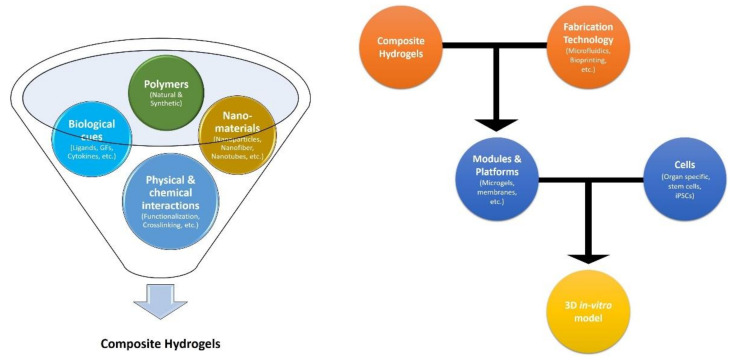
Schematic representation of various factors and components in the development of hydrogels and hydrogel-based in vitro 3D model.

**Figure 3 ijms-23-02662-f003:**
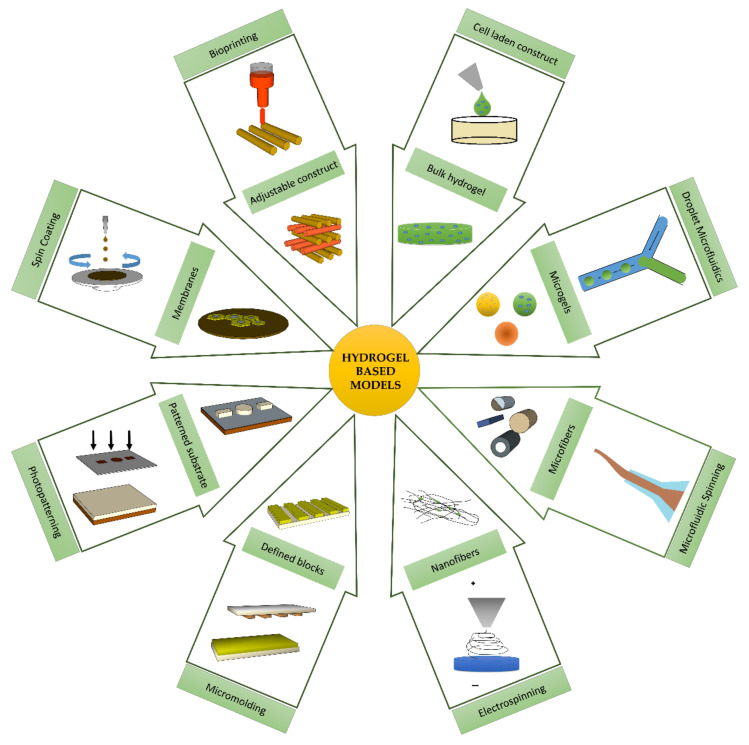
Schematic figure of existing fabrication technologies for designing various hydrogel units/platforms.

**Figure 4 ijms-23-02662-f004:**
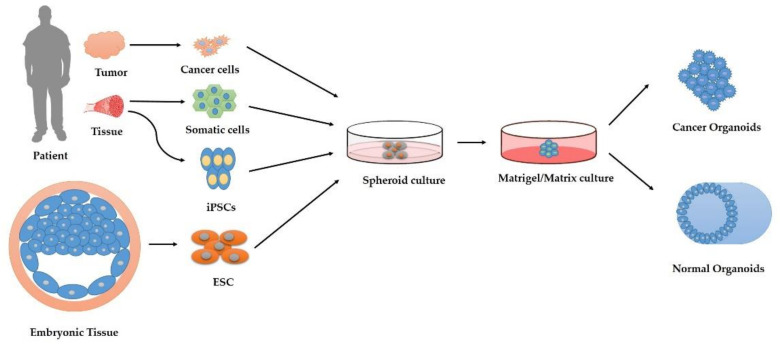
The development of organoids from primary cells, stem cells, and cancer cells. Embryonic stem cells (ESC) from embryonic tissue and iPSCs derived from adult primary tissue first experience direct differentiation into a floating spheroid culture. The culture is subsequently transferred into a matrix-based hydrogel with a specific medium to initiate the formation of organoids. Somatic stem cells can be obtained from patient tissue, which can be further cultured in a 3D medium to develop it into organoids. Cancer cells from tumor tissue can be developed into tumoroids in a specific 3D culture.

**Figure 5 ijms-23-02662-f005:**
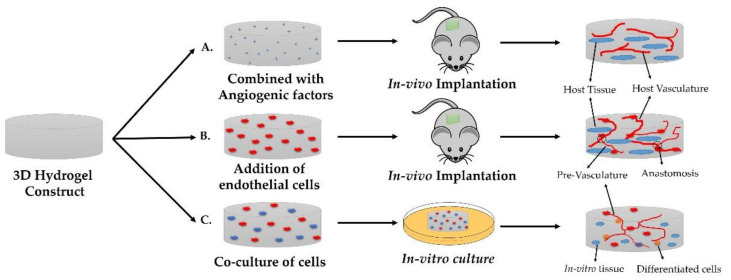
Vascularization strategies in 3D tissue construct. A. Hydrogel construct incorporated with angiogenic factors. Host vasculature developed in 3D construct upon in vivo implantation. B. Formation of pre-vascularization and anastomosis upon in vivo implantation of endothelial cell culture hydrogels. C. Co-culture study is more suitable in in vitro culture system.

**Table 1 ijms-23-02662-t001:** Comparison of cell behavior under 2D and 3D culture conditions.

S. No.	Cell-Specific Features	2D Culture	3D Culture	Ref.
1	Morphology and architecture of cells	Flat and extended morphology with poor cell architecture	A rich architecture with round and contracted morphology	[8,9,10]
2	Migration of cells	Cells migration is fast and directional	Migration of cells in all directions with slow and restricted motility	[11,12]
3	Proliferation of cells	High proliferation rate	Relatively low proliferation compared to 2D culture	[11,13]
4	Interaction with surroundings	Limited interaction with cells and ECMs	Cells can interact with their surrounding in all directions	[14]
5	Polarization of cells	Partly polarized	Full polarization	[11]
6	Intracellular metabolism	High metabolic rate	Relatively low rate of intracellular metabolism	[12]
7	Diffusion of fluids and cell signaling	Limited perfusion of fluids with asymmetric metabolite diffusion and cell signaling	Three-dimensional fluid perfusion and symmetric diffusion of metabolites	[15,16]
8	ECM remodeling	Poor or near absent	Close to mimicking the in-vivo ECM remodeling	[13]
9	Cell viability against cytotoxic agents	High loss of cells	High cell survival rate	[17]
10	Cell death/Apoptosis	Induced apoptosis	Tissue-like apoptosis	[12,18]

**Table 2 ijms-23-02662-t002:** Survey of hybrid hydrogel composition for in vitro 3D cell culture.

Hydrogel Composition	Gelation/Fabrication Method Used	Distinctive Features	Findings	Reference
Natural–Natural Polymers				
Alginate–Matrigel	Ionic crosslinking with Ca++ ions	Increased hydrogel stiffness	Progression of the normal mammary epithelium into malignant cells	[109]
Alginate–Marine Collagen–Agarose	Self-assembly	Increased cytocompatibility	High yield in multicellular spheroids	[110]
Alginate–Gelatin	Pre-crosslinking with calcium chloride	Similar mechanical properties as crosslinked alginate but superior cytocompatibility	Prospective bio-ink provides a good means for myoregenerative applications	[111]
Chitosan–Gelatin	Ionic interaction to form polyelectrolyte complexes	High shape fidelity and good biocompatibility	Bioprinting of skin fibroblasts	[112]
Gelatin–Silk Fibroin	Enzymatic and physical (sonication) crosslinking	Superior mechanical strength, tunable degradability, and improved multilineage differentiation ability	Site-specific bioprinting of progenitor cells and differentiation	[113]
GelMA–Gellan gum MA	Photo-crosslinking	Achieved similar viscoelasticity to native cartilage	Enhanced viability and growth for human articular chondrocytes	[114]
Thiolated Gelatin–Vinyl sulfonated HA	Click chemistry (-SH with C=C)	High viability and proliferation capability	Bone MSC in vitro differentiation toward chondrocytes	[115]
Amphiphilic peptides–keratin	Self-co-assembly	Integration of self-assembly and bioprinting. The process directed molecular assembly and nanomaterials into ordered structures with various sizes and geometries	Introduced a 3D bioprinting platform encapsulating cells in the pericellular environment	[116]
Natural–Synthetic Polymers				
PEG–Keratin	thiol-norbornene “click” reaction	Highly tunable mechanical properties and long-term stability	Suitable for various microfabrication techniques (e.g., micropatterning, wet spinning) for fabrication of 3D cell-laden tissue constructs	[117]
PCL–RGD peptides	Acrylic acid (AAC) grafting by γ irradiation and crosslinking by 1-ethyl-3-(3-dimethylaminopropyl)carbodiimide/N-hydroxysuccinimide (EDC/NHS)	Provides a fibrous anchorage site in the 3D hydrogel environment	Mesenchymal stem cells show remarkable spreading properties with augmented viability and differentiation	[118]
PCL–Alginate	Interfacial bonding	Composite mimics the microarchitecture and mechanical properties of soft tissue	Subcutaneous implantation shows infiltration of pro-regenerative macrophages and leads to gradual remodeling and replacement of the composites with vascularized soft tissue	[119]
PCL–PEG–heparin	Melt-electrospinning writing combined with additive manufacturing	Fibrous networks exhibited mechanical anisotropy, viscoelasticity, and morphology similar to native cartilage tissue	In vitro neo-cartilage formation	[120]
PCL–GelMA–Alginate	Melt-electrospinning writing	Stiffness and elasticity were similar to those of the native articular cartilage tissue	Embedded human chondrocytes retain their round morphology and are responsive to in vitro physiological loading regime	[121]
Sodium alginate/PLGA microspheres–HSP27–TAT peptide	Crosslinking with calcium sulfate followed by physical incorporation of microspheres	Porous microsphere enabled sustained release of HSP27-TAT hybrid system for over two weeks	Sustained delivery of HSP27-TAT reduced the infarcted site and improved the end-systolic volume in the heart	[122]
Chondroitin sulfate–PEG	Enzymatic crosslinking (Transglutaminase factor XIII)	Modular design allows the facile incorporation of additional signaling element	Tunable matrix with BMP2 binding and sustained release allows enhanced proliferation of MSCs and differentiation toward osteogenic lineage	[123]
Chitosan/HA–PLGA microsphere	Reversible Schiff’s base reaction	The linking of vancomycin with PLGA microspheres enabled the hydrogel system to inhibit bacterial growth	Vascular endothelial growth factor (VEGF) encapsulation to the PLGA microspheres accelerated the growth and proliferation of endothelial cells and increased angiogenesis, thereby promoting management of non-healing wounds	[124]
Hydrogel linked with biological factors				
PEGDA–RGD peptide	Two-photon laser scanning photolithography	Generates microscale patterns with control over the spatial distribution of biomolecules	Human dermal fibroblasts cultured in fibrin clusters of hybrid hydrogel underwent a guided 3D migration	[125]
Star PEG–desulfated Heparin–VEGF	EDC/NHS crosslinking	The presence of heparin provides anticoagulant activity, while sustained release of VEGF ensures the growth of endothelial cells	In vitro tube formation of human umbilical vein endothelial cells (HUVEC) and promotion of wound healing in genetically diabetic mice	[126]
GelMA–VEGF	EDC/NHS crosslinking; Extrusion-based direct-writing bioprinting	Inside the bioprinted 3D construct, a central cylinder of GelMA was printed to allow the formation of perfusable blood vessels	Co-culture of MSCs enabled the formation and stabilization of endothelial cells. VEGF-linked hydrogel induced differentiation of MSCs toward osteogenesis	[127]
GelMA–BMP2–TGF-β1	Bioprinting; Photo-crosslinking	Established an anisotropic biomimetic fibrocartilage microenvironment by bioprinting a nanoliter droplet encapsulating MSCs, BMP-2, and TGF-β1	Genomic expression study findings indicate differentiation of MSCs and simultaneous upregulation of osteogenic and chondrogenic factors during the in vitro culture on the model 3D construct	[128]
Collagen–fibrin–VEGF	Culture plate coated with a nebulized layer of sodium bicarbonate; Bioprinting	An artificial neural tissue construct was fabricated by bioprinting collagen-laden neural stem cells and VEGF-embedded fibrin gel	A sustained release of VEGF was found from bioprinted fibrin gel, which enhanced the migration and proliferation of neural stem cells	[129]
Matrigel–Gelatin microparticles–VEGF	Glutaraldehyde crosslinking	The construct of gelatin microparticles is suitable to generate sustained release profiles of bioactive VEGF	In vitro study shows real-time migration of endothelial progenitor cells and reveals enhanced in vivo vasculogenic capacity	[130]
Hydrogels incorporated with nanomaterials				
Dopamine–Folic acid–ZnO–quantum dot	Crosslinking by transition metal ions to form metal–ligand coordination	The hydrogels provide greater antibacterial efficacy when illuminated at 660 and 808 nm (generate ROS and heat).	The hydrogels release zinc ions over two weeks, provide a sustained antimicrobial effect against *S. aureus* and *E. coli*, and promote fibroblast growth	[131]
Alginate–SiO_2_ nanofibers	Ionic crosslinking through Al^3+^	The method helps the nanofibrous hydrogels retain a large amount of water, which helps in producing desirable shapes at a larger scale	Results of zero Poisson’s ratio, memory of shape, injectability, and conductivity provide insight into the development of future multifunctional hydrogels	[132]
Poly hydroxyethylmethacrylate (pHEMA)–multiwalled carbon nanotube (MWCNT)	Polymerization at 4 °C	With incorporated MWCNT, the construct has a more porous structure with better elastic modulus and electrical conductivity	The in vitro study shows the viability of neuroblastoma cells and that they help conduct electricity, indicating that the construct is more suitable as a nerve conduit	[133]
HA–dopamine–rGO	EDC/NHS crosslinking; Oxidative coupling of catechol groups by H_2_O_2_/HRP as the initiator	Multifunctional, including tissue adhesiveness, antibacterial and antioxidant ability, and good mechanical properties	Shows significant skin regeneration capacity with enhanced vascularization; promoted as an excellent wound-dressing hydrogel	[134]
Quaternized Chitosan–benzaldehyde terminated F127	Crosslinking between Schiff base bond and PF127 micelle	The dressings system showed good stretchability, similar mechanics to human native skin, and rapid self-healing	In an in vivo study with a full-thickness skin defect model, the hydrogel showed deposition of collagen with upregulated VEGF, which led to accelerated wound healing	[135]
Alginate–PLA nanofibers	Ultrasonication	Comparable compressive characteristics to native alginate hydrogels with better cytocompatibility	The nanofiber-blended bio-ink allowed the enhancement of adipose-derived stem cells proliferation, with the presence of collagen and proteoglycans indicating chondrogenic differentiation	[136]
Alginate–Nanofibrillated cellulose	90 mM CaCl_2_	The bio-ink shear-thinning behaviors enable printing of 2D grid structures as well as complex 3D soft tissue-like constructs	Bioprinted human chondrocytes show over 80% viability 7 days post-culture, demonstrating the potential of the bio-ink for 3D bioprinting	[137]

**Table 3 ijms-23-02662-t003:** Types of bioprinting methods and their features.

Bioprinting Method	Working Method	Advantage	Limitation
Micro-extrusion method	Most common method. Physical force is used to distribute biomaterial and cells at a specific location through a nozzle.	Can print heterogeneous and complex structures	Low resolution printing
Droplet-based method	A controlled volume of cell suspension hydrogel is printed at the desired location. The print volume can be controlled via a magnetic field, an electric heating nozzle, and piezoelectric or acoustic actuators.	Much more accurate resolution than micro-extrusion printing	Difficult to print large-scale biological structures
Laser-assisted printing	Biological structures are printed by laser-guided front transfer. The solidification method uses a laser-induced photo-polymerization using UV, infrared, or visible light.	Prints at the highest resolution owing to the laser interference	Low cellular viability

## Data Availability

The data presented in this study are available on request from the corresponding author.

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
