# Peer review of "Engineering Hydrogels for the Development of Three-Dimensional In Vitro Models"

_ijms, 2022, doi:10.3390/ijms23052662_

Round 1

Reviewer 1 Report

In the Review paper by Maji et al., the authors aimed to summarize the different aspects of using hydrogels for the development of 3D in vitro models, including the different types of hydrogels, different manufacturing techniques, and the different organs.

While this review paper includes a lot of relevant information and covers a wide field of research, the provided information is organized inefficiently, thus delivering a shallow survey of the literature accompanied by a poor discussion. As a consequence, some of the information or comments are repeated many times along the text in very similar sentences. Since the major task of a review paper is to deliver broad and extensive data in a well-organized and interesting manner, as well as to discuss it, this manuscript did not manage to accomplish its task. Overall, following its very promising title, the manuscript was a very long disappointment.

Additional and more specific comments:

  • Major English editing is required. The entire text should be revised in terms of singular/plural, grammatical tense, etc.
  • Non-scientific terms should be avoided (e.g. “cell friendly”, and many more)
  • Figure 2. Shows a very poor representation of “designing and development of Hydrogels in engineering of in-vitro three-dimensional model”.
  • Several examples and issues are only relevant for tissue engineering and not for in vitro models. While in some cases mentioning tissue engineering as the basis to a certain approach that was further adapted for in vitro models is relevant, in most cases the tissue engineering issues are just mixed in the text in a confusing or distracting manner. (Discussing immunogenicity of materials and ear bioprinting are just two examples).
  • When focusing on the fabrication technologies, the main subject of the review—hydrogels—is not discussed in terms of their particular contribution and the choice of appropriate materials for each technology.
  • A main discussion that is essential in such a review paper is how, in the authors’ eyes, should in vitro models be evaluated. Clear criteria should be suggested.
  • In vitro models applying mechanical and/or electrical stimuli should be discussed in terms of the relevant suitable hydrogels.
  • Decellularized ECM is mentioned as an irrelevant hydrogel due to low reproducibility. Nevertheless, the processing of dECM into a hydrogel enables overcoming its poor reproducibility through the incorporation of tissues from several animals into the same batch of a product. Furthermore, this material exhibits unique tissue-specific advantages for 3D models that were emphasized in both academic publications (such as reference # 316, PMID: 31075518, PMID: 26480475, PMID: 32337805, PMID: 34661396) and commercial products based on dECM (such as Sigma’s tissue-specific dECM Gel Hydrogel Kits).
  • When discussing microgels production, other technologies were not taken into consideration, such as electrostatic-driven and air-driven microgel generation, as well as emulsion methods.

Reviewer 2 Report

I recommend acceptance for this review article.

Author Response

Response: We appreciate reviewer’s warm work earnestly.

Reviewer 3 Report

This is an excellent review paper describing recent progress on hydrogels for the development of three-dimensional in vitro model. The structure of the paper is well organized and the references are up-to-date. In addition, "Challenges and Future Perspective" section is a very nice guidance for researchers in this field. I recommend publication of this paper with a little modifications. I have a few minor comments:

1) line 81, what is "water products"? I did not understand the meaning.

2) section 4.3, there are several values (0.51 kPa, 1.41 kPa, and 6 kPa) describing the mechanical properties of hydrogels. Are these storage modulus of the gel? They should be clarified.

3) (Figure 3) in line 843 and (Figure 4) in line 922 should be Figure 4 and 5, respectively.

Author Response

Reviewer 3

Comments and Suggestions for Authors

This is an excellent review paper describing recent progress on hydrogels for the development of three-dimensional in vitro model. The structure of the paper is well organized and the references are up-to-date. In addition, "Challenges and Future Perspective" section is a very nice guidance for researchers in this field. I recommend publication of this paper with a little modifications.

I have a few minor comments:

Point 1: line 81, what are "water products"? I did not understand the meaning. 

Response 1: We apologize for the confusion. The term “water products” has been corrected to “waste product”.

Point 2: Section 4.3, there are several values (0.51 kPa, 1.41 kPa, and 6 kPa) describing the mechanical properties of hydrogels. Are these storage modulus of the gel? They should be clarified.

Response 2: Thank you very much for your question.  The values 0.51 KPa and 1.41 KPa represent hydrogel’s compressive modulus whereas the value 6 KPa represents the elastic modulus. The same has been clarified in the manuscript.

[Sentences related to the response]: Page- 22, Section- 4.3, Line- 807-810; 816-819

In a  study by Wu et al., the compressive modulus of hydrogels were prepared in lower (0.51 KPa) and upper (1.41 KPa) stiffness ranges  of native brain tissue to encapsulate neural progenitor cells [219].

The observation indicates the maximum differentiation efficiency and contractility percentage of cardiac organoid when presented the substrate similar to the viscoelasticity of native cardiac muscle (elastic modulus - 6 KPa).

Point 3: (Figure 3) in line 843 and (Figure 4) in line 922 should be Figure 4 and 5, respectively.

Response 3: Thank you very much for the observation. We have corrected the figure number in the manuscript.

Round 2

Reviewer 1 Report

  • Major English editing is required, from the very first sentence and the entire manuscript. There are still expressions such as "they presents", "hydrogels works", etc. 
  • While the manuscript was extensively revised, some of the revisions correspond to the comments, while others are irrelevant.
  • There is still confusion between text about tissue engineering and text about in vitro models.
  • While this manuscript brings together a lot of information from many publications, this information is served as a literature survey rather than a critical review. 
